# Inverse Density as an Inverse Problem: the Fredholm Equation Approach

**Qichao Que, Mikhail Belkin**
Department of Computer Science and Engineering
The Ohio State University
{que,mbelkin}@cse.ohio-state.edu

## Abstract

We address the problem of estimating the ratio $\frac{q}{p}$ where $p$ is a density function and $q$ is another density, or, more generally an arbitrary function. Knowing or approximating this ratio is needed in various problems of inference and integration often referred to as *importance sampling* in statistical inference. It is also closely related to the problem of *covariate shift* in transfer learning. Our approach is based on reformulating the problem of estimating the ratio as an inverse problem in terms of an integral operator corresponding to a kernel, known as the Fredholm problem of the first kind. This formulation, combined with the techniques of regularization leads to a principled framework for constructing algorithms and for analyzing them theoretically. The resulting family of algorithms (FIRE, for Fredholm Inverse Regularized Estimator) is flexible, simple and easy to implement. We provide detailed theoretical analysis including concentration bounds and convergence rates for the Gaussian kernel for densities defined on $\mathbb{R}^d$ and smooth $d$-dimensional sub-manifolds of the Euclidean space.

Model selection for unsupervised or semi-supervised inference is generally a difficult problem. It turns out that in the density ratio estimation setting, when samples from both distributions are available, simple completely unsupervised model selection methods are available. We call this mechanism CD-CV for Cross-Density Cross-Validation. We show encouraging experimental results including applications to classification within the covariate shift framework.

## 1 Introduction

In this paper we address the problem of estimating the ratio of two functions, $\frac{q(x)}{p(x)}$ where $p$ is given by a sample and $q(x)$ is either a known function or another probability density function given by a sample. This estimation problem arises naturally when one attempts to integrate a function with respect to one density, given its values on a sample obtained from another distribution. Recently there have been a significant amount of work on estimating the density ratio (also known as the importance function) from sampled data, e.g., [6, 10, 9, 22, 2]. Many of these papers consider this problem in the context of *covariate shift* assumption [19] or the so-called *selection bias* [27]. The approach taken in our paper is based on reformulating the density ratio estimation as an integral equation, known as the Fredholm equation of the first kind, and solving it using the tools of regularization and Reproducing Kernel Hilbert Spaces. That allows us to develop simple and flexible algorithms for density ratio estimation within the popular kernel learning framework. The connection to the classical operator theory setting makes it easier to apply the standard tools of spectral and Fourier analysis to obtain theoretical results.

We start with the following simple equality underlying the *importance sampling* method:

$$E_q(h(x)) = \int h(x)q(x)dx = \int h(x)\frac{q(x)}{p(x)}p(x)dx = E_p\left(h(x)\frac{q(x)}{p(x)}\right) \qquad (1)$$

By replacing the function $h(x)$ with a kernel $k(x, y)$, we obtain

$$\mathcal{K}_p \frac{q}{p}(x) := \int k(x, y) \frac{q(y)}{p(y)} p(y) dy = \int k(x, y) q(y) dy := \mathcal{K}_q \mathbf{1}(x). \tag{2}$$

Thinking of the function $\frac{q(x)}{p(x)}$ as an unknown quantity and assuming that the right hand side is known this becomes a Fredholm integral equation. Note that the right-hand side can be estimated given a sample from $q$ while the operator on the left can be estimated using a sample from $p$.

To push this idea further, suppose $k_t(x, y)$ is a "local" kernel, (e.g., the Gaussian, $k_t(x, y) = \frac{1}{(2\pi t)^{d/2}} e^{-\frac{\|x-y\|^2}{2t}}$) such that $\int_{\mathbb{R}^d} k_t(x, y) dx = 1$. When we use $\delta$-kernels, like Gaussian, and $f$ satisfies some smoothness conditions, we have $\int_{\mathbb{R}^d} k_t(x, y) f(x) dx = f(y) + O(t)$ (see [24], Ch. 1). Thus we get another (approximate) integral equality:

$$\mathcal{K}_{t,p} \frac{q}{p}(y) := \int_{\mathbb{R}^d} k_t(x, y) \frac{q(x)}{p(x)} p(x) dx \approx q(y). \tag{3}$$

It becomes an integral equation for $\frac{q(x)}{p(x)}$, assuming that $q$ is known or can be approximated.

We address these inverse problems by formulating them within the classical framework of Tiknonov-Philips regularization with the penalty term corresponding to the norm of the function in the Reproducing Kernel Hilbert Space $\mathcal{H}$ with kernel $k_{\mathcal{H}}$ used in many machine learning algorithms.

[Type I]: $\frac{q}{p} \approx \arg\min_{f \in \mathcal{H}} \|\mathcal{K}_p f - \mathcal{K}_q \mathbf{1}(x)\|_{L_{2,p}}^2 + \lambda \|f\|_{\mathcal{H}}^2$  [III]: $\frac{q}{p} \approx \arg\min_{f \in \mathcal{H}} \|\mathcal{K}_{t,p} f - q\|_{L_{2,p}}^2 + \lambda \|f\|_{\mathcal{H}}^2$

Importantly, given a sample $x_1, \ldots, x_n$ from $p$, the integral operator $\mathcal{K}_p f$ applied to a function $f$ can be approximated by the corresponding discrete sum $\mathcal{K}_p f(x) \approx \frac{1}{n} \sum_i f(x_i) K(x_i, x)$, while $L_{2,p}$ norm is approximated by an average: $\|f\|_{L_{2,p}}^2 \approx \frac{1}{n} \sum_i f(x_i)^2$. Of course, the same holds for a sample from $q$. We see that the Type I formulation is useful when $q$ is a density and samples from both $p$ and $q$ are available, while the Type II is useful, when the values of $q$ (which does not have to be a density function at all[1]) are known at the data points sampled from $p$.

Since all of these involve only function evaluations at the sample points, an application of the usual representer theorem for Reproducing Kernel Hilbert Spaces, leads to simple, explicit and easily implementable algorithms, representing the solution of the optimization problem as linear combinations of the kernels over the points of the sample $\sum_i \alpha_i k_{\mathcal{H}}(x_i, x)$ (see Section 2). We call the resulting algorithms FIRE for Fredholm Inverse Regularized Estimator.

**Remark: Other norms and loss functions.** Norms and loss functions other that $L_{2,p}$ can also be used in our setting as long as they can be approximated from a sample using function evaluations.
**1.** Perhaps, the most interesting is $L_{2,q}$ norm available in the Type I setting, when a sample from the probability distribution $q$ is available. In fact, given a sample from both $p$ and $q$ we can use the combined empirical norm $\gamma \| \cdot \|_{L_{2,p}} + (1 - \gamma) \| \cdot \|_{L_{2,q}}$. Optimization using those norms leads to some interesting kernel algorithms described in Section 2. We note that the solution is still a linear combination of kernel functions centered on the sample from $p$ and can still be written explicitly.
**2.** In Type I formulation, if the kernels $k(x, y)$ and $k_{\mathcal{H}}(x, y)$ coincide, it is possible to use the RKHS norm $\| \cdot \|_{\mathcal{H}}$ instead of $L_{2,p}$. This formulation (see Section 2) also yields an explicit formula and is related to the Kernel Mean Matching [9] , although with a different optimization procedure.

Since we are dealing with a classical inverse problem for integral operators, our formulation allows for theoretical analysis using the methods of spectral theory. In Section 3 we present concentration and error bounds as well as convergence rates for our algorithms when data are sampled from a distribution defined in $\mathbb{R}^d$, a domain in $\mathbb{R}^d$ with boundary or a compact $d$-dimensional sub-manifold of a Euclidean space $\mathbb{R}^N$ for the case of the Gaussian kernel.

In Section 4 we introduce a unsupervised method, referred as CD-CV (for cross-density cross-validation) for model selection and discuss the experimental results on several data sets comparing our method FIRE with the available alternatives, Kernel Mean Matching (KMM) [9] and LSIF [10] as well as the base-line thresholded inverse kernel density estimator[2] (TIKDE) and importance sampling (when available).

We summarize the contributions of the paper as follows:

**1.** We provide a formulation of estimating the density ratio (importance function) as a classical inverse problem, known as the Fredholm equation, establishing a connections to the methods of classical analysis. The underlying idea is to "linearize" the properties of the density by studying an associated integral operator.

**2.** To solve the resulting inverse problems we apply regularization with an RKHS norm penalty. This provides a flexible and principled framework, with a variety of different norms and regularization techniques available. It separates the underlying inverse problem from the necessary regularization and leads to a family of very simple and direct algorithms within the kernel learning framework in machine learning.

**3.** Using the techniques of spectral analysis and concentration, we provide a detailed theoretical analysis for the case of the Gaussian kernel, for Euclidean case as well as for distributions supported on a sub-manifold. We prove error bounds and as well as the convergence rates.

**4.** We also propose a completely unsupervised technique, CD-CV, for cross-validating the parameters of our algorithm and demonstrate its usefulness, thus addressing in our setting one of the most thorny issues in unsupervised/semi-supervised learning. We evaluate and compare our methods on several different data sets and in various settings and demonstrate strong performance and better computational efficiency compared to the alternatives.

**Related work.** Recently the problem of density ratio estimation has received significant attention due in part to the increased interest in transfer learning [15] and, in particular to the form of transfer learning known as covariate shift [19]. To give a brief summary, given the feature space $X$ and the label space $Y$, two probability distributions $p$ and $q$ on $X \times Y$ satisfy the covariate assumption if for all $x, y$, $p(y|x) = q(y|x)$. It is easy to see that training a classifier to minimize the error for $q$, given a sample from $p$ requires estimating the ratio of the marginal distributions $\frac{q_X(x)}{p_X(x)}$. The work on covariate shift, density ratio estimation and related settings includes [27, 2, 6, 10, 22, 9, 23, 14, 7].

The algorithm most closely related to ours is Kernel Mean Matching [9]. It is based on the equation: $E_q(\Phi(x)) = E_p(\frac{q}{p}\Phi(x))$, where $\Phi$ is the feature map corresponding to an RKHS $\mathcal{H}$. It is rewritten as an optimization problem $\frac{q(x)}{p(x)} \approx \arg\min_{\beta \in L_2, \beta(x)>0, E_p(\beta)=1} \|E_q(\Phi(x)) - E_p(\beta(x)\Phi(x))\|_{\mathcal{H}}$. The quantity on the right can be estimated given a sample from $p$ and a sample from $q$ and the minimization becomes a quadratic optimization problem over the values of $\beta$ at the points sampled from $p$. Writing down the feature map explicitly, i.e., recalling that $\Phi(x) = K_{\mathcal{H}}(x, \cdot)$, we see that the equality $E_q(\Phi(x)) = E_p(\frac{q}{p}\Phi(x))$ is equivalent to the integral equation Eq. 2 considered as an identity in the Hilbert space $\mathcal{H}$. Thus the problem of KMM can be viewed within our setting Type I (see the Remark 2 in the introduction), with a RKHS norm but a different optimization algorithm.

However, while the KMM optimization problem uses the RKHS norm, the weight function $\beta$ itself is not in the RKHS. Thus, unlike most other algorithms in the RKHS framework (in particular, FIRE), the empirical optimization problem does not have a natural out-of-sample extension. Also, since there is no regularizing term, the problem is less stable (see Section 4 for some experimental comparisons) and the theoretical analysis is harder (however, see [6] and the recent paper [26] for some nice theoretical analysis of KMM in certain settings).

Another related recent algorithm is Least Squares Importance Sampling (LSIF) [10], which attempts to estimate the density ratio by choosing a parametric linear family of functions and choosing a function from this family to minimize the $L_{2,p}$ distance to the density ratio. A similar setting with the Kullback-Leibler distance (KLIEP) was proposed in [23]. This has an advantage of a natural out-of-sample extension property. We note that our method for unsupervised parameter selection in Section 4 is related to their ideas. However, in our case the set of test functions does not need to form a good basis since no approximation is required.

We note that our methods are closely related to a large body of work on kernel methods in machine learning and statistical estimation (e.g., [21, 17, 16]). Many of these algorithms can be interpreted as inverse problems, e.g., [3, 20] in the Tikhonov regularization or other regularization frameworks. In particular, we note interesting methods for density estimation proposed in [12] and estimating the support of density through spectral regularization in [4], as well as robust density estimation using RKHS formulations [11] and conditional density [8]. We also note the connections of the methods in this paper to properties of density-dependent operators in classification and clustering [25, 18, 1]. Among those works that provide theoretical analysis of algorithms for estimating density ratios,

[14] establishes minimax rates for likelihood ratio estimation. Another recent theoretical analysis of KMM in [26] contains bounds for the output of the corresponding integral operators.

## 2 Settings and Algorithms

**Settings and objects.** We start by introducing objects and function spaces important for our development. As usual, the norm in space of square-integrable functions with respect to a measure $\rho$, is defined as follows: $L_{2,\rho} = \left\{ f : \int_\Omega |f(x)|^2 d\rho < \infty \right\}$. This is a Hilbert space with the inner product defined in the usual way by $\langle f, g \rangle_{2,\rho} = \int_\Omega f(x)g(x)d\rho$. Given a kernel $k(x,y)$ we define the operator $\mathcal{K}_\rho$: $\mathcal{K}_\rho f(y) := \int_\Omega k(x,y)f(x)d\rho(x)$. We will use the notation $\mathcal{K}_{t,\rho}$ to explicitly refer to the parameter of the kernel function $k_t(x,y)$, when it is a $\delta$-family. If the function $k(x,y)$ is symmetric and positive definite, then there is a corresponding *Reproducing Kernel Hilbert space* (RKHS) $\mathcal{H}$. We recall the key property of the kernel $k_\mathcal{H}$: for any $f \in \mathcal{H}$, $\langle f, k_\mathcal{H}(x,\cdot) \rangle_\mathcal{H} = f(x)$. The Representer Theorem allows us to write solutions to various optimization problems over $\mathcal{H}$ in terms of linear combinations of kernels supported on sample points (see [21] for an in-depth discussion or the RKHS theory and the issues related to learning). Given a sample $x_1, \ldots, x_n$ from $p$, one can approximate the $L_{2,p}$ norm of a sufficiently smooth function $f$ by $\|f\|_{2,p}^2 \approx \frac{1}{n} \sum_i |f(x_i)|^2$, and similarly, the integral operator $K_p f(x) \approx \frac{1}{n} \sum_i k(x_i, x)f(x_i)$. These approximate equalities can be made precise by using appropriate concentration inequalities.

**The FIRE Algorithms.** As discussed in the introduction, the starting point for our development is the two integral equalities,

[I]: $\mathcal{K}_p \dfrac{q}{p}(\cdot) = \int k(\cdot, y) \dfrac{q(y)}{p(y)} dp(y) = \mathcal{K}_q \mathbf{1}(\cdot)$   [II]: $\mathcal{K}_{t,p} \dfrac{q}{p}(\cdot) = \int k_t(\cdot, y) \dfrac{q(y)}{p(y)} dp(y) = q(\cdot) + o(1)$ 
$$\tag{4}$$

Notice that in the Type I setting, the kernel does not have to be in a $\delta$-family. For example, a linear kernel is admissible. Type II setting comes from the fact $\mathcal{K}_{t,q} f(x) \approx f(x)p(x) + O(t)$ for a "$\delta$-function-like" kernel and we keep $t$ in the notation in that case. Assuming that either $\mathcal{K}_q \mathbf{1}$ or $q$ are (approximately) known (Type I and II settings, respectively) equalities in Eqs. 4 become integral equations for $\frac{p}{q}$, known as Fredholm equations of the first kind. To estimate $\frac{p}{q}$, we need to obtain an approximation to the solution which (a) can be obtained computationally from sampled data, (b) is stable with respect to sampling and other perturbation of the input function, (c) can be analyzed using the standard machinery of functional analysis.

To provide a framework for solving these inverse problems, we apply the classical techniques of regularization combined with the RKHS norm popular in machine learning. In particular a simple formulation of Type I using Tikhonov regularization, ([5], Ch. 5), with the $L_{2,p}$ norm is as follows:

$$[\text{Type I}]: \quad f_\lambda^{\text{I}} = \arg\min_{f \in \mathcal{H}} \|\mathcal{K}_p f - \mathcal{K}_q \mathbf{1}\|_{2,p}^2 + \lambda \|f\|_\mathcal{H}^2 \tag{5}$$

Here $\mathcal{H}$ is an appropriate Reproducing Kernel Hilbert Space. Similarly Type II can be solved by

$$[\text{Type II}]: \quad f_\lambda^{\text{II}} = \arg\min_{f \in \mathcal{H}} \|\mathcal{K}_{t,p} f - q\|_{2,p}^2 + \lambda \|f\|_\mathcal{H}^2 \tag{6}$$

We will now discuss the empirical versions of these equations and the resulting algorithms.

**Type I setting. Algorithm for $L_{2,p}$ norm.** Given an iid sample from $p$, $\boldsymbol{z}_p = \{x_i\}_{i=1}^n$ and an iid sample from $q$, $\boldsymbol{z}_q = \{x_j'\}_{j=1}^m$ ($\boldsymbol{z}$ for the combined sample), we can approximate the integral operators $\mathcal{K}_p$ and $\mathcal{K}_q$ by $K_{\boldsymbol{z}_p} f(x) = \frac{1}{n} \sum_{x_i \in \boldsymbol{z}_p} k(x_i, x)f(x_i)$ and $K_{\boldsymbol{z}_q} f(x) = \frac{1}{m} \sum_{x_i' \in \boldsymbol{z}_q} k(x_i', x)f(x_i')$. Thus the empirical version of Eq. 5 becomes

$$f_{\lambda,\boldsymbol{z}}^{\text{I}} = \arg\min_{f \in \mathcal{H}} \frac{1}{n} \sum_{x_i \in \boldsymbol{z}_p} ((\mathcal{K}_{\boldsymbol{z}_p} f)(x_i) - (\mathcal{K}_{\boldsymbol{z}_q} \mathbf{1})(x_i))^2 + \lambda \|f\|_\mathcal{H}^2 \tag{7}$$

The first term of the optimization problem involves only evaluations of the function $f$ at the points of the sample. From Representer Theorem and matrix manipulation, we obtain the following:

$$f_{\lambda,\boldsymbol{z}}^{\text{I}}(x) = \sum_{x_i \in \boldsymbol{z}_p} k_\mathcal{H}(x_i, x)v_i \text{ and } \boldsymbol{v} = \left(K_{p,p}^2 K_\mathcal{H} + n\lambda I\right)^{-1} K_{p,p} K_{p,q} \mathbf{1}_{\boldsymbol{z}_q}. \tag{8}$$

where the kernel matrices are defined as follows: $(K_{p,p})_{ij} = \frac{1}{n} k(x_i, x_j)$, $(K_\mathcal{H})_{ij} = k_\mathcal{H}(x_i, x_j)$ for $x_i, x_j \in \boldsymbol{z}_p$ and $K_{p,q}$ is defined as $(K_{p,q})_{ij} = \frac{1}{m} k(x_i, x_j')$ for $x_i \in \boldsymbol{z}_p$ and $x_j' \in \boldsymbol{z}_q$.

If $K_{\mathcal{H}}$ and $K_{p,p}$ are the same kernel we simply have: $\boldsymbol{v} = \frac{1}{n}\left(K_{p,p}^3 + \lambda I\right)^{-1} K_{p,p}K_{p,q}\mathbf{1}_{\boldsymbol{z}_q}$.

**Algorithms for $\gamma L_{2,p} + (1-\gamma)L_{2,q}$ norm.** Depending on the setting, we may want to minimize the error of the estimate over the probability distribution $p, q$ or over some linear combination of these. A significant potential benefit of using a linear combination is that both samples can be used at the same time in the loss function. First we state the continuous version of the problem:

$$f_\lambda^* = \arg\min_{f\in\mathcal{H}} \gamma\|\mathcal{K}_p f - \mathcal{K}_q\mathbf{1}\|_{2,p}^2 + (1-\gamma)\|\mathcal{K}_p f - \mathcal{K}_q\mathbf{1}\|_{2,q}^2 + \lambda\|f\|_{\mathcal{H}}^2 \qquad (9)$$

Given a sample from $p$, $\boldsymbol{z}_p = \{x_1, x_2, \ldots, x_n\}$ and a sample from $q$, $\boldsymbol{z}_q = \{x_1', x_2', \ldots, x_m'\}$ we obtain an empirical version of the Eq. 9: $f_{\lambda,\boldsymbol{z}}^*(x) =$

$$\arg\min_{f\in\mathcal{H}} \frac{\gamma}{n}\sum_{x_i\in\boldsymbol{z}_p}\left(\mathcal{K}_{\boldsymbol{z}_p}f(x_i) - \mathcal{K}_{\boldsymbol{z}_q}\mathbf{1}(x_i^p)\right)^2 + \frac{1-\gamma}{m}\sum_{x_i'\in\boldsymbol{z}_q}\left((\mathcal{K}_{\boldsymbol{z}_p}f)(x_i') - (\mathcal{K}_{\boldsymbol{z}_q}\mathbf{1})(x_i')\right)^2 + \lambda\|f\|_H^2$$

From the Representer Theorem $f_{\lambda,\boldsymbol{z}}^*(x) = \sum_{x_i\in\boldsymbol{z}_p} v_i k_{\mathcal{H}}(x_i, x)$ $\qquad \boldsymbol{v} = (K + n\lambda I)^{-1} K_1\mathbf{1}_{\boldsymbol{z}_q}$

$$K = \left(\frac{\gamma}{n}(K_{p,p})^2 + \frac{1-\gamma}{m}K_{q,p}^T K_{q,p}\right)K_{\mathcal{H}} \text{ and } K_1 = \left(\frac{\gamma}{n}K_{p,p}K_{p,q} + \frac{1-\gamma}{m}K_{q,p}^T K_{q,q}\right)$$

where $(K_{p,p})_{ij} = \frac{1}{n}k(x_i, x_j)$, $(K_{\mathcal{H}})_{ij} = k_{\mathcal{H}}(x_i, x_j)$ for $x_i, x_j \in \boldsymbol{z}_p$, and $(K_{p,q})_{ij} = \frac{1}{m}k(x_i, x_j')$ and $(K_{q,p})_{ji} = \frac{1}{n}k(x_j', x_i)$ for $x_i \in \boldsymbol{z}_p, x_j' \in \boldsymbol{z}_q$. Despite the loss function combining both samples, the solution is still a summation of kernels over the points in the sample from $p$.

**Algorithms for the RKHS norm.** In addition to using the RKHS norm for regularization norm, we can also use it as a loss function: $f_\lambda^* = \arg\min_{f\in\mathcal{H}} \|\mathcal{K}_p f - \mathcal{K}_q\mathbf{1}\|_{\mathcal{H}'}^2 + \lambda\|f\|_{\mathcal{H}}^2$ Here the Hilbert space $\mathcal{H}'$ must correspond to the kernel $k$ and can potentially be different from the space $\mathcal{H}$ used for regularization. Note that this formulation is only applicable in the Type I setting since it requires the function $q$ to belong to the RKHS $\mathcal{H}'$. Given two samples $\boldsymbol{z}_p, \boldsymbol{z}_q$, it is easy to write down the empirical version of this problem, leading to the following formula:

$$f_{\lambda,\boldsymbol{z}}^*(x) = \sum_{x_i\in\boldsymbol{z}_p} v_i k_{\mathcal{H}}(x_i, x) \qquad \boldsymbol{v} = (K_{p,p}K_{\mathcal{H}} + n\lambda I)^{-1} K_{p,q}\mathbf{1}_{\boldsymbol{z}_q}. \qquad (10)$$

The result is somewhat similar to our Type I formulation with the $L_{2,p}$ norm. We note the connection between this formulation of using the RKHS norm as a loss function and the KMM algorithm [9]. When the kernels $K$ and $K_{\mathcal{H}}$ are the same, Eq. 10 can be viewed as a regularized version of KMM (with a different optimization procedure).

**Type II setting.** In Type II setting we assume that we have a sample $\boldsymbol{z} = \{x_i\}_{i=1}^n$ drawn from $p$ and that we know the function values $q(x_i)$ at the points of the sample. Replacing the norm and the integral operator with their empirical versions, we obtain the following optimization problem:

$$f_{\lambda,\boldsymbol{z}}^{\mathrm{II}} = \arg\min_{f\in\mathcal{H}} \frac{1}{n}\sum_{x_i\in\boldsymbol{z}}(\mathcal{K}_{t,\boldsymbol{z}_p}f(x_i) - q(x_i))^2 + \lambda\|f\|_{\mathcal{H}}^2 \qquad (11)$$

As before, using the Representer Theorem we obtain an analytical formula for the solution:

$$f_{\lambda,\boldsymbol{z}}^{\mathrm{II}}(x) = \sum_{x_i\in\boldsymbol{z}} k_{\mathcal{H}}(x_i, x)v_i \text{ where } \boldsymbol{v} = \left(K^2 K_{\mathcal{H}} + n\lambda I\right)^{-1} K\mathbf{q}.$$

where the kernel matrix $K$ is defined by $K_{ij} = \frac{1}{n}k_t(x_i, x_j)$, $(K_{\mathcal{H}})_{ij} = k_{\mathcal{H}}(x_i, x_j)$ and $\mathbf{q}_i = q(x_i)$.

**Comparison of type I and type II settings.**
**1.** In Type II setting $q$ does not have to be a density function (i.e., non-negative and integrate to one).
**2.** Eq. 7 of the Type I setting cannot be easily solved in the absence of a sample $\boldsymbol{z}_q$ from $q$, since estimating $\mathcal{K}_q$ requires either sampling from $q$ (if it is a density) or estimating the integral in some other way, which may be difficult in high dimension but perhaps of interest in certain low-dimensional application domains.
**3.** There are a number of problems (e.g., many problems involving MCMC) where $q(x)$ is known explicitly (possibly up to a multiplicative constant), while sampling from $q$ is expensive or even impossible computationally [13].
**4.** Unlike Eq. 5, Eq. 6 has an error term depending on the kernel. For example, in the important case of the Gaussian kernel, the error is of the order $O(t)$, where $t$ is the variance of Gaussian.
**5.** Several norms are available in the Type I setting, but only the $L_{2,p}$ norm is available for Type II.

# 3 Theoretical analysis: bounds and convergence rates for Gaussian Kernels

In this section, we state our main results on bounds and convergence rates for our algorithm based on Tikhonov regularization with a Gaussian kernel. We consider both Type I and Type II settings for the Euclidean and manifold cases and make a remark on the Euclidean domains with boundary. To simplify the theoretical development, the integral operator and the RKHS $\mathcal{H}$ will correspond to the same Gaussian kernel $k_t(x, y)$. The proofs will be found in the supplemental material.

**Assumptions:** The set $\Omega$, where the density function $p$ is defined, could be one of the following: (1) the whole $\mathbb{R}^d$; (2) a compact smooth Riemannian sub-manifold $\mathcal{M}$ of $d$-dimension in $\mathbb{R}^n$. We also need $p(x) < \Gamma, q(x) < \Gamma$ for any $x \in \Omega$ and that $\frac{q}{p}, \frac{q}{p^2}$ are in Sobolev space $W_2^2(\Omega)$.

**Theorem 1. ( Type I setting.)** *Let $p$ and $q$ be two density functions on $\Omega$. Given $n$ points, $\boldsymbol{z}_p = \{x_1, x_2, \ldots, x_n\}$, i.i.d. sampled from $p$ and $m$ points, $\boldsymbol{z}_q = \{x'_1, x'_2, \ldots, x'_m\}$, i.i.d. sampled from $q$, and for small enough $t$, for the solution to the optimization problem in (7), with confidence at least $1 - 2e^{-\tau}$, we have*

*(1) If the domain $\Omega$ is $\mathbb{R}^d$, for some constants $C_1, C_2, C_3$ independent of $t, \lambda$.*

$$\left\| f_{\lambda,\boldsymbol{z}}^I - \frac{q}{p} \right\|_{2,p} \le C_1 t + C_2 \lambda^{\frac{1}{2}} + C_3 \frac{\sqrt{\tau}}{\lambda t^{d/2}} \left( \frac{1}{\sqrt{m}} + \frac{1}{\lambda^{1/6}\sqrt{n}} \right) \tag{12}$$

*(2) If the domain $\Omega$ is a compact sub-manifold without boundary of $d$ dimension, for some $0 < \varepsilon < 1$, $C_1, C_2, C_3$ independent of $t, \lambda$.*

$$\left\| f_{\lambda,\boldsymbol{z}}^I - \frac{q}{p} \right\|_{2,p} \le C_1 t^{1-\varepsilon} + C_2 \lambda^{\frac{1}{2}} + C_3 \frac{\sqrt{\tau}}{\lambda t^{d/2}} \left( \frac{1}{\sqrt{m}} + \frac{1}{\lambda^{1/6}\sqrt{n}} \right) \tag{13}$$

**Corollary 2. ( Type I setting.)** *Assuming $m > \lambda^{1/3} n$, with confidence at least $1 - 2e^{-\tau}$, when (1) $\Omega = \mathbb{R}^d$, (2) $\Omega$ is a $d$-dimensional sub-manifold of a Euclidean space, we have*

$$(1) \left\| f_{\lambda,\boldsymbol{z}}^I - \frac{q}{p} \right\|_{2,p}^2 = O\left(\sqrt{\tau} n^{-\frac{1}{3.5+d/2}}\right) (2) \left\| f_{\lambda,\boldsymbol{z}}^I - \frac{q}{p} \right\|_{2,p}^2 = O\left(\sqrt{\tau} n^{-\frac{1}{3.5(1-\varepsilon)+d/2}}\right) \forall \varepsilon \in (0,1)$$

**Theorem 3. ( Type II setting.)** *Let $p$ be a density function on $\Omega$ and $q$ be a function satisfying the assumptions. Given $n$ points $\boldsymbol{z} = \{x_1, x_2, \ldots, x_n\}$ sampled i.i.d. from $p$, and for sufficiently small $t$, for the solution to the optimization problem in (11), with confidence at least $1 - 2e^{-\tau}$, we have*

*(1) If the domain $\Omega$ is $\mathbb{R}^d$,*

$$\left\| f_{\lambda,\boldsymbol{z}}^{II} - \frac{q}{p} \right\|_{2,p} \le C_1 t + C_2 \lambda^{\frac{1}{2}} + C_3 \lambda^{-\frac{1}{3}} \|\mathcal{K}_{t,q}\mathbf{1} - q\|_{2,p} + C_4 \frac{\sqrt{\tau}}{\lambda^{3/2} t^{d/2}\sqrt{n}}, \tag{14}$$

*where $C_1, C_2, C_3, C_4$ are constants independent of $t, \lambda$. Moreover, $\|\mathcal{K}_{t,q}\mathbf{1} - q\|_{2,p} = O(t)$.*

*(2) If $\Omega$ is a $d$-dimensional sub-manifold of a Euclidean space, for any $0 < \varepsilon < 1$*

$$\left\| f_{\lambda,\boldsymbol{z}}^{II} - \frac{q}{p} \right\|_{2,p} \le C_1 t^{1-\varepsilon} + C_2 \lambda^{\frac{1}{2}} + C_3 \lambda^{-\frac{1}{3}} \|\mathcal{K}_{t,q}\mathbf{1} - q\|_{2,p} + C_4 \frac{\sqrt{\tau}}{\lambda^{3/2} t^{d/2}\sqrt{n}}, \tag{15}$$

*where $C_1, C_2, C_3, C_4$ are independent of $t, \lambda$. Moreover, $\|\mathcal{K}_{t,q}\mathbf{1} - q\|_{2,p} = O(t^{1-\eta}), \forall \eta > 0$.*

**Corollary 4. ( Type II setting.)** *With confidence at least $1 - 2e^{-\tau}$, when (1) $\Omega = \mathbb{R}^d$, (2) $\Omega$ is a $d$-dimensional sub-manifold of a Euclidean space, we have*

$$(1) \left\| f_{\lambda,\boldsymbol{z}}^{II} - \frac{q}{p} \right\|_{2,p}^2 = O\left(\sqrt{\tau} n^{-\frac{1}{4+\frac{5}{6}d}}\right) (2) \left\| f_{\lambda,\boldsymbol{z}}^{II} - \frac{q}{p} \right\|_{2,p}^2 = O\left(\sqrt{\tau} n^{-\frac{1-\eta}{4-4\eta+\frac{5}{6}d}}\right) \forall \eta \in (0,1)$$

# 4 Model Selection and Experiments

We describe an unsupervised technique for parameter selection, Cross-Density Cross-Validation (CD-CV) based on a performance measure unique to our setting. We proceed to evaluate our method.

**The setting.** In our experiments, we have $X^p = \{x_1^p, \ldots, x_n^p\}$ and $X^q = \{x_1^q, \ldots, x_m^q\}$. The goal is to estimate $\frac{q}{p}$, assuming that $X^p, X^q$ are i.i.d. sampled from $p, q$ respectively. Note that

learning $\frac{q}{p}$ is unsupervised and our algorithms typically have two parameters: the kernel width $t$ and regularization parameter $\lambda$.

**Performance Measures and CD-CV Model Selection.** We describe a set of performance measures used for parameter selection. For a given function $u$, we have the following importance sampling equality (Eq. 1): $\mathbb{E}_p(u(x)) = \mathbb{E}_q\left(u(x)\frac{p(x)}{q(x)}\right)$. If $f$ is an approximation of the true ratio $\frac{q}{p}$, and $X^p, X^q$ are samples from $p, q$ respectively, we will have the following approximation to the previous equation: $\frac{1}{n}\sum_{i=1}^{n} u(x_i^p)f(x_i^p) \approx \frac{1}{m}\sum_{j=1}^{m} u(x_j^q)$. So after obtaining an estimate $f$ of the ratio, we can validate it using the following performance measure:

$$J_{CD}(f; X^p, X^q, U) = \frac{1}{F}\sum_{l=1}^{F}\left(\sum_{i=1}^{n} u_l(x_i^p)f(x_i^p) - \sum_{j=1}^{m} u_l(x_j^q)\right)^2 \qquad (16)$$

where $U = \{u_1, \ldots, u_F\}$ is a collection of test functions. Using this performance measure allows various cross-validation procedures to be used for parameter selection. We note that this way to measure error is related to the LSIF [10] and KLIEP [23] algorithms. However, there a similar measure is used to construct an approximation to the ratio $\frac{q}{p}$ using functions $u_1, \ldots, u_F$ as a basis. In our setting, we can use test functions (e.g., linear functions) which are poorly suited as a basis for approximating the density ratio.

We will use the following two families of test functions for parameter selection: (1) Sets of random linear functions $u_i(x) = \beta^T x$ where $\beta \sim N(0, Id)$; (2) Sets of random half-space indicator functions, $u_i(x) = \mathbf{1}_{\beta^T x > 0}$.

**Procedures for parameter selection.** The performance is optimized using five-fold cross-validation by splitting the data set into two parts $X_{train}^p$ and $X_{train}^q$ for training and $X_{cv}^p$ and $X_{cv}^q$ for validation. The range we use for kernel width $t$ is $(t_0, 2t_0, \ldots, 2^9 t_0)$, where $t_0$ is the average distance of the 10 nearest neighbors. The range for regularization parameter $\lambda$ is $(1e-5, 1e-6, \ldots, 1e-10)$.

**Data sets and Resampling** We use two datasets, CPUsmall and Kin8nm, for regression; and USPS handwritten digits for classification. And we draw the first 500 or 1000 points from the original data set as $X^p$. To obtain $X^q$, the following two ways of resampling, using the features or the label information, are used (along the lines of those in [6]).

Given a set of data with labels $\{(x_1, y_1), (x_2, y_2), \ldots, (x_n, y_n)\}$ and denoting $P_i$ the probability of $i$'th instance being chosen, we resample as follows:

(1) *Resampling using features (labels $y_i$ are not used).* $P_i = \frac{e^{(a\langle x_i, e_1\rangle - b)/\sigma_v}}{1 + e^{(a\langle x_i, e_1\rangle - b)/\sigma_v}}$, where $a, b$ are the resampling parameters, $e_1$ is the first principal component, and $\sigma_v$ is the standard deviation of the projections to $e_1$. This resampling method will be denoted by PCA$(a, b)$.

(2) *Resampling using labels.* $P_i = \begin{cases} 1 & y_i \in L_q \\ 0 & \text{Otherwise.} \end{cases}$ where $y_i \in L = \{1, 2, \ldots, k\}$ and $L_q$ is a subset of the whole label set $L$. It only applies to binary problems obtained by aggregating different classes in multi-class setting.

**Testing the FIRE algorithm.** In the first experiment, we test our method for selecting parameters by focusing on the error $J_{CD}(f; X^p, X^q, U)$ in Eq. 16 for different function classes $U$. Parameters are chosen using a family of functions $U_1$, while the performance of the parameter is measured using an independent function family $U_2$. This measure is important because in practice the functions we are interested in may not be the ones chosen for validation.

We use the USPS data sets for this experiment. As a basis for comparison we use TIKDE (Thresholded Inverse Kernel Density Estimator). TIKDE estimates $\hat{p}$ and $\hat{q}$ respectively using Kernel Density Estimation (KDE), and assigns $\hat{p}(x) = \alpha$ to any $x$ satisfying $\hat{p}(x) < \alpha$. TIKDE then outputs $\hat{q}/\hat{p}$. We note that chosen threshold $\alpha$ is key to reasonable performance. One issue of this heuristic is that it could underestimate at the region with high density ratio, due to the uniform thresholding. We also compare our methods to LSIF [10]. In these experiments we do not compare with KMM as out-of-sample extension is necessary for fair comparison.

Table 1 shows the average errors of various methods, defined in Eq. 16 on held-out set $X^{err}$ over 5 trials. We use different validation functions $f^{cv}$(Columns) and error-measuring functions $f^{err}$(Row). $N$ is the number of random functions used for validation. The error-measuring function families $U_2$ are as follows: (1) Linear(L.): random linear functions $f(x) = \beta^T x$ where $\beta \sim N(0, Id)$; (2) Half-

space(H.S.): Sets of random half-space indicator functions, $f(x) = \mathbf{1}_{\beta^T x}$; (3) Kernel(K.): random linear combinations of kernel functions centered at training data, $f(x) = \gamma^T K$ where $\gamma \sim N(0, Id)$ and $K_{ij} = k(x_i, x_j)$ for $x_i$ from training set; (4) Kernel indicator(K.I.) functions $f(x) = \mathbf{1}_{g(x)>0}$, where $g$ is as in (3).

Table 1: USPS data set with resampling using PCA$(5, \sigma_v)$ with $|X^p| = 500$, $|X^q| = 1371$. Around 400 in $X^p$ and 700 in $X^q$ are used in 5-fold CV, the rest are held-out for computing the error.

| | | Linear | | Half-Spaces | | | | Linear | | Half-Spaces | |
|---|---|---|---|---|---|---|---|---|---|---|---|
| | N | 50 | 200 | 50 | 200 | | N | 50 | 200 | 50 | 200 |
| L. | TIKDE | 10.9 | 10.9 | 10.9 | 10.9 | K. | TIKDE | 4.7 | 4.7 | 4.7 | 4.7 |
| | LSIF | 14.1 | 14.1 | 26.8 | 28.2 | | LSIF | 16.1 | 16.1 | 15.6 | 13.8 |
| | FIRE$_p$ | **3.6** | 3.7 | 5.5 | 6.3 | | FIRE$_p$ | 1.2 | **1.1** | 2.8 | 3.6 |
| | FIRE$_{p,q}$ | 4.7 | 4.7 | 7.4 | 6.8 | | FIRE$_{p,q}$ | 2.1 | 2.0 | 4.2 | 2.6 |
| | FIRE$_q$ | 5.9 | 6.2 | 9.3 | 9.3 | | FIRE$_q$ | 5.2 | 4.3 | 6.1 | 6.1 |
| H.S. | TIKDE | 2.6 | 2.6 | 2.6 | 2.6 | K.I. | TIKDE | 4.2 | 4.2 | 4.2 | 4.2 |
| | LSIF | 3.9 | 3.9 | 3.7 | 3.9 | | LSIF | 4.4 | 4.4 | 5.3 | 4.4 |
| | FIRE$_p$ | 1.0 | **0.9** | 1.0 | 1.2 | | FIRE$_p$ | 0.9 | 0.7 | 1.2 | 1.1 |
| | FIRE$_{p,q}$ | **0.9** | 1.0 | 1.4 | 1.1 | | FIRE$_{p,q}$ | **0.6** | **0.6** | 1.9 | 1.1 |
| | FIRE$_q$ | 1.2 | 1.4 | 1.6 | 1.6 | | FIRE$_q$ | 1.2 | 0.9 | 2.2 | 2.2 |

**Supervised Learning: Regression and Classification.** We compare our FIRE algorithm with several other methods in regression and classification tasks. We consider the situation where part of the data set $X^p$ are labeled and all of $X^q$ are unlabeled. We use weighted ordinary least-square for regression and weighted linear SVM for classification.

**Regression.** Square loss function is used for regression. The performance is measured using normalized square loss, $\sum_{i=1}^{n} \frac{(\hat{y}_i - y_i)^2}{\text{Var}(\hat{y} - y)}$. $X^q$ is resampled using PCA resampler, described before. L is for Linear, and HS is for Half-Space function families for parameter selection.

Table 2: Mean normalized square loss on the CPUsmall and Kin8nm. $|X^p| = 1000$, $|X^q| = 2000$.

| | CPUsmall, resampled by PCA$(5, \sigma_v)$ | | | | | | Kin8nm, resampled by PCA$(1, \sigma_v)$ | | | | | |
|---|---|---|---|---|---|---|---|---|---|---|---|---|
| No. of Labeled | 100 | | 200 | | 500 | | 100 | | 200 | | 500 | |
| Weights | L | HS | L | HS | L | HS | L | HS | L | HS | L | HS |
| OLS | .74 | | .50 | | .83 | | .59 | | .55 | | 0.54 | |
| TIKDE | .38 | .36 | .30 | .29 | .28 | .28 | .57 | .57 | .55 | .55 | .53 | .53 |
| KMM | 1.86 | 1.86 | 1.9 | 1.9 | 2.5 | 2.5 | .58 | .58 | .55 | .55 | **.52** | **.52** |
| LSIF | .39 | .39 | .31 | .31 | .33 | .33 | .57 | **.56** | **.54** | **.54** | **.52** | **.52** |
| FIRE$_p$ | .33 | .33 | .29 | .29 | **.27** | **.27** | .57 | **.56** | .55 | **.54** | **.52** | **.52** |
| FIRE$_{p,q}$ | .33 | .33 | .29 | .29 | **.27** | **.27** | **.56** | **.56** | .55 | **.54** | **.52** | **.52** |
| FIRE$_q$ | **.32** | .33 | **.28** | .29 | **.27** | **.27** | **.56** | **.56** | .55 | **.54** | **.52** | **.52** |

**Classification.** Weighted linear SVM. Percentage of incorrectly labeled test set instances.

Table 3: Average error on USPS with +1 class= $\{0 - 4\}$, −1 class= $\{5 - 9\}$ and $|X^p| = 1000$ and $|X^q| = 2000$. Left half of the table uses resampling PCA$(5, \sigma_v)$, where $\sigma_v$. Right half shows resampling based on **Label** information.

| | PCA$(5, \sigma_v)$ | | | | | | $L = \{\{0 - 4\}, \{5 - 9\}\}, L' = \{0, 1, 5, 6\}$ | | | | | |
|---|---|---|---|---|---|---|---|---|---|---|---|---|
| No. of Labeled | 100 | | 200 | | 500 | | 100 | | 200 | | 500 | |
| Weights | L | HS | L | HS | L | HS | L | HS | L | HS | L | HS |
| SVM | 10.2 | | 8.1 | | 5.7 | | 18.6 | | 16.4 | | 12.9 | |
| TIKDE | 9.4 | 9.4 | 7.2 | 7.2 | 4.9 | 4.9 | 18.5 | 18.5 | 16.4 | 16.4 | 12.4 | 12.4 |
| KMM | 8.1 | 8.1 | 5.9 | 5.9 | 4.7 | 4.7 | **17.5** | **17.5** | **13.5** | **13.5** | **10.3** | **10.3** |
| LSIF | 9.5 | 10.2 | 7.3 | 8.1 | 5.0 | 5.7 | 18.5 | 18.5 | 16.2 | 16.3 | 12.2 | 12.2 |
| FIRE$_p$ | 8.9 | 6.8 | 5.3 | 5.0 | **4.1** | **4.1** | 17.9 | 18.4 | 16.1 | 16.1 | 11.5 | 12.0 |
| FIRE$_{p,q}$ | 7.0 | 7.0 | 5.1 | 5.1 | **4.1** | **4.1** | 18.0 | 18.5 | 16.1 | 16.2 | 11.6 | 12.0 |
| FIRE$_q$ | **5.5** | 7.3 | **4.8** | 5.4 | **4.1** | 4.4 | 18.3 | 18.4 | 16.0 | 16.2 | 11.8 | 12.0 |

**Acknowledgements.** The work was partially supported by NSF Grants IIS 0643916, IIS 1117707.

## Footnotes

[1]This could be useful in sampling procedures, when the normalizing coefficients are hard to estimate.

[2]The standard kernel density estimator for $q$ divided by a thresholded kernel density estimator for $p$.

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
