[Supplementary Material]

# Supplementary Material: Inverse Density as an Inverse Problem

**Qichao Que, Mikhail Belkin**
Department of Computer Science and Engineering
The Ohio State University

## Abstract

We address the problem of estimating the ratio $\frac{q}{p}$ where $p$ is a density function and $q$ is another density, or, more generally an arbitrary function. Knowing or approximating this ratio is needed in various problems of inference and integration, in particular, when one needs to average a function with respect to one probability distribution, given a sample from another. It is often referred as *importance sampling* in statistical inference and is also closely related to the problem of *covariate shift* in transfer learning as well as to various MCMC methods. Our approach is based on reformulating the problem of estimating the ratio as an inverse problem in terms of an integral operator corresponding to a kernel, and thus reducing it to an integral equation, known as the Fredholm problem of the first kind. This formulation, combined with the techniques of regularization and kernel methods, leads to a principled kernel-based framework for constructing algorithms and for analyzing them theoretically. The resulting family of algorithms (FIRE, for Fredholm Inverse Regularized Estimator) is flexible, simple and easy to implement. We provide detailed theoretical analysis including concentration bounds and convergence rates for the Gaussian kernel for densities defined on $\mathbb{R}^d$ and smooth $d$-dimensional sub-manifolds of the Euclidean space.

Model selection for unsupervised or semi-supervised inference is generally a difficult problem. Interestingly, it turns out that in the density ratio estimation setting, when samples from both distributions are available, there are simple completely unsupervised methods for choosing parameters. We call this model selection mechanism CD-CV for Cross-Density Cross-Validation. Finally, we show encouraging experimental results including applications to classification within the covariate shift framework.

## 1   Introduction

Density estimation is one of the best-studied and most useful problems in statistical inference. The question is to estimate the probability density function $p(x)$ from a sample $x_1, \ldots, x_n$. There is a rich literature on the subject (e.g., see the review [12]), particularly, dealing with a class of non-parametric kernel estimators going back to the work of Parzen [22].

In this paper we address the related problem of estimating the ratio of two functions, $\frac{q(x)}{p(x)}$ where $p$ is given by a sample and $q(x)$ is either a known function or another probability density function given by a sample. We note that estimating such ratio is necessary when one attempts to integrate a function with respect to one density, given its values on a sample obtained from another distribution. This is typical when the process generating the data is different from the averaging problem we wish to address. To give a very simple practical example of such a situation, consider a cleaning robot equipped with a dirt sensor. We would like to know how well the robot performs cleaning, however, the probability density of the robot location $p(x)$ depends on the program and is clearly not uniform.

To obtain the cleaning quality, we need to average the sensor readings with respect to the uniform density over the floor rather than the location distribution, which requires estimating the inverse $\frac{1}{p}$ (here $q(x)$ it the constant function 1).

An important class of applications for density ratios relates to various Markov Chain Monte Carlo (MCMC) integration techniques used in various applications, in particular, in many tasks of Bayesian inference. It is often hard to sample directly from the desirable probability distribution but it may be possible to construct an approximation which is easier to sample from. The class of techniques related to the *importance sampling* (see, e.g., [17]) deals with this problem by using a ratio of two densities (which is typically assumed to be known in that literature).

Recently there have been a significant amount of work on estimating the density ratio (also known as te importance function) from sampled data, e.g., [9, 14, 11, 30, 4]. Many of these papers consider this problem in the context of *covariate shift* assumption [26] or the so-called *selection bias* [38]. Our Fredholm Inverse Regularized Estimator (FIRE) framework introduces a very general and flexible approach to this problem which leads to more efficient algorithms design, provides very competitive experimental results and makes possible theoretical analysis in terms of the sample complexity and convergence rates.

We will provide a more detailed discussion of these and other related papers and connections to our work in Section 2, where we also discuss how the Kernel Mean Matching algorithm [9, 11] can be viewed within our framework.

The approach taken in our paper is based on reformulating the density ratio estimation as an integral equation, known as the Fredholm equation of the first kind (in the classical one-dimensional case), and solving it using the tools of regularization and Reproducing Kernel Hilbert Spaces. That allows us to develop simple and flexible algorithms for density ratio estimation within the popular kernel learning framework. In addition the integral operator approach separates estimation and regularization problems, thus allowing us to address certain settings where the existing methods are not applicable. The connection to the classical operator theory setting makes it easier to apply the standard tools of spectral analysis to obtain theoretical results.

We will now briefly outline the main idea of this paper. We start with the following simple equality underlying the *importance sampling* method:

$$E_q(h(x)) = \int h(x)q(x)dx = \int h(x)\frac{q(x)}{p(x)}p(x)dx = E_p\left(h(x)\frac{q(x)}{p(x)}\right) \qquad (1)$$

By replacing the function $h(x)$ with a kernel $k(x,y)$, we obtain

$$\mathcal{K}_p\frac{q}{p}(x) := \int k(x,y)\frac{q(y)}{p(y)}p(y)dy = \int k(x,y)q(y)dy := \mathcal{K}_q\mathbf{1}(x). \qquad (2)$$

Thinking of the function $\frac{q(x)}{p(x)}$ as an unknown quantity and assuming that the right hand side is known this becomes a Fredholm integral equation. Note that the right-hand side can be estimated given a sample from $q$ while the operator on the left can be estimated using a sample from $p$.

To push this idea further, suppose $k_t(x,y)$ is a "local" kernel, (e.g., the Gaussian, $k_t(x,y) = \frac{1}{(2\pi t)^{d/2}}e^{-\frac{\|x-y\|^2}{2t}}$) such that $\int_{\mathbb{R}^d} k_t(x,y)dx = 1$. When we use $\delta$-kernels, like Gaussian, and $f$ satisfies some smoothness conditions, we have $\int_{\mathbb{R}^d} k_t(x,y)f(x)dx = f(y) + O(t)$ (see [33], Ch. 1). Thus we get another (approximate) integral equality:

$$\mathcal{K}_{t,p}\frac{q}{p}(y) := \int_{\mathbb{R}^d} k_t(x,y)\frac{q(x)}{p(x)}p(x)dx \approx q(y). \qquad (3)$$

It becomes an integral equation for $\frac{q(x)}{p(x)}$, assuming that $q$ is known or can be approximated.

We address these inverse problems by formulating them within the classical framework[1] of Tiknonov-Philips regularization with the penalty term corresponding to the norm of the function

in the Reproducing Kernel Hilbert Space $\mathcal{H}$ with kernel $k_{\mathcal{H}}$ used in many machine learning algorithms.

[Type I]: $\frac{q}{p} \approx \arg\min_{f \in \mathcal{H}} \|\mathcal{K}_p f - \mathcal{K}_q \mathbf{1}\|_{L_{2,p}}^2 + \lambda \|f\|_{\mathcal{H}}^2$   [Type II]: $\frac{q}{p} \approx \arg\min_{f \in \mathcal{H}} \|\mathcal{K}_{t,p} f - q\|_{L_{2,p}}^2 + \lambda \|f\|_{\mathcal{H}}^2$

Importantly, given a sample $x_1, \ldots, x_n$ from $p$, the integral operator $\mathcal{K}_p f$ applied to a function $f$ can be approximated by the corresponding discrete sum $\mathcal{K}_p f(x) \approx \frac{1}{n} \sum_i f(x_i) k(x_i, x)$, while $L_{2,p}$ norm is approximated by an average: $\|f\|_{L_{2,p}}^2 \approx \frac{1}{n} \sum_i f(x_i)^2$. Of course, the same holds for a sample from $q$.

Thus, we see that the Type I formulation is useful when $q$ is a density and samples from both $p$ and $q$ are available, while the Type II is useful, when the values of $q$ (which does not have to be a density function at all[2]) are known at the data points sampled from $p$.

Since all of these involve only function evaluations at the sample points, by an application of the usual representer theorem for Reproducing Kernel Hilbert Spaces, both Type I and II formulations lead to simple, explicit and easily implementable algorithms, representing the solution of the optimization problem as linear combinations of the kernels over the points of the sample $\sum_i \alpha_i k_{\mathcal{H}}(x_i, x)$ (see Section 3). We call the resulting algorithms FIRE for Fredholm Inverse Regularized Estimator.

Some remarks would be useful at this point.
**Remark 1: Other norms and loss functions.** Norms and loss functions other that $L_{2,p}$ can also be used in our setting as long as they can be approximated from a sample using function evaluations.

- Perhaps, the most interesting is the norm $L_{2,q}$ norm available in the Type I setting, when a sample from the probability distribution $q$ is available. In fact, given a sample from both $p$ and $q$ we can use the combined empirical norm $\gamma \|\cdot\|_{L_{2,p}} + (1-\gamma)\|\cdot\|_{L_{2,q}}$. Optimization using those norms leads to some interesting looking kernel algorithms described in Section 3. We note that the solution is still a linear combination of kernel functions centered on the sample from $p$ and can still be written explicitly.

- In the Type I formulation, if the kernels $k(x,y)$ and $k_{\mathcal{H}}(x,y)$ coincide, it is possible to use the RKHS norm $\|\cdot\|_{\mathcal{H}}$ instead of $\|\cdot\|_{L_{2,p}}$. This formulation (see Section 3) also yields an explicit formula and is related to the Kernel Mean Matching algorithm [11] (see the discussion in Section 2), although with a different optimization procedure. We note that the solution in our framework has a natural out-of-sample extension, which becomes important for proper parameter selection.

- Other norms/loss functions, e.g., $L_{1,p}, L_{1,q}$, $\epsilon$-insensitive loss from the SVM regression, etc., can also be used in our framework as long as they can be approximated from a sample using function evaluations. We note that some of these may have advantages in terms of the sparsity of the resulting solution. On the other hand, a standard advantage of using the square norm is the ease of cross-validation with respect to the parameter $\lambda$.

**Remark 2: Other regularization methods.** Several regularization methods other than Tikhonov-Philips regularization are available. We will briefly discuss the *spectral cut-off regularization* and its potential advantages in Section 3. We note that other methods, such as early stopping (e.g., [36, 1]) can be used and may have computational advantages.

**Remark 3.** We note that an intermediate **"Type 1.5"** formulation is also available. Specifically, for two "$\delta$-kernels" $K$ and $K'$, we have $\mathcal{K}_p \frac{q}{p} \approx \mathcal{K}_q' \mathbf{1}$, thus using two different kernels in the Type I formulation

$$\frac{q}{p} \approx \arg\min_{f \in \mathcal{H}} \|\mathcal{K}_p f - \mathcal{K}_q' \mathbf{1}\|_{L_{2,p}}^2 + \lambda \|f\|_{\mathcal{H}}^2 \qquad (4)$$

The ability to use kernels with different bandwidth for $p$ and $q$ may be potentially important in practice, especially when the samples from $p$ and $q$ have very different cardinality. The resulting

algorithms for this setting are described in in Section 3. Of course, the previous two remarks apply to this setting as well.

Since we are dealing with a classical inverse problem for integral operators, our formulation allows for theoretical analysis using the methods of spectral theory. In Section 4 we prove concentration and error bounds as well as convergence rates for our algorithms when data are sampled from a distribution defined in $\mathbb{R}^d$, a domain in $\mathbb{R}^d$ with boundary or a compact $d$-dimensional sub-manifold of a Euclidean space $\mathbb{R}^N$ for the case of the Gaussian kernel.

It is interesting to note that unlike the usual density estimation problem the width of the kernel does not need to go to zero for convergence. However, it is necessary if we want a polynomial convergence rate. This is related to the exponential decay of eigenvalues for the Gaussian kernel.

Finally, in Section 6, we introduce a unsupervised method, referred as CD-CV (for Cross-density cross-validation), for model selection, and discuss the experimental results on several data sets comparing our method FIRE with the available alternatives, Kernel Mean Matching (KMM) [11] and LSIF [14] as well as the base-line thresholded inverse kernel density estimator[3] (TIKDE) and importance sampling (when available).

We summarize the contributions of the paper as follows:

1. We provide a formulation of estimating the density ratio (importance function) as a classical inverse problem, known as the Fredholm equation, establishing a connections to the methods of classical analysis. The underlying idea is to "linearize" the properties of the density by studying an associated integral operator.

2. To solve the resulting inverse problems we apply regularization with an RKHS norm penalty. This provides a flexible and principled framework, with a variety of different norms and regularization techniques available. It separates the underlying inverse problem from the necessary regularization and leads to a family of very simple and direct algorithms within the kernel learning framework in machine learning. We call the resulting algorithms FIRE for Fredholm Inverse Regularized Estimator.

3. Using the techniques of spectral analysis and concentration, we provide a detailed theoretical analysis for the case of the Gaussian kernel, for Euclidean case as well as distributions supported on a sub-manifold. We prove error bounds and as well as the convergence rates (as far as we know, it is the first convergence rate analysis for density ratio estimation). We also comment on other kernels and potential extensions of our analysis.

4. We also propose a completely unsupervised technique, CD-CV, for cross-validating the parameters of our algorithm and demonstrate its usefulness, thus addressing in our setting one of the most thorny issues in unsupervised/semi-supervised learning. We evaluate and compare our methods on several different data sets and in various settings and demonstrate strong performance and better computational efficiency compared to the alternatives.

5. Finally, our framework allows us to address several different settings related to a number of problems in areas from covariate shift classification in transfer learning to importance sampling in MCMC to geometry estimation and numerical integration. Some of these connections are explored in this paper and some we hope to address in the future work.

## 2   Related work

The problem of density estimation has a long history in classical statistical literature and a rich variety of methods are available [12]. However, as far as we know the problem of estimating the inverse density or density ratio from a sample has not been studied extensively until quite recently. Some of the related older work includes density estimation for inverse problems [8] and the literature on deconvolution, e.g., [5].

In the last few years the problem of density ratio estimation has received significant attention due in part to the increased interest in transfer learning [21] and, in particular to the form of transfer

learning known as covariate shift [26]. To give a brief summary, given the feature space $X$ and the label space $Y$, two probability distributions $p$ and $q$ on $X \times Y$ satisfy the covariate assumption if for all $x, y$, $p(y|x) = q(y|x)$. It is easy to see that training a classifier to minimize the error for $q$, given a sample from $p$ requires estimating the ratio of the marginal distributions $\frac{q_X(x)}{p_X(x)}$. Some of the work on covariate shift, ratio density estimation and other closely related settings includes [38, 4, 9, 14, 30, 11, 31, 13, 20]

The algorithm most closely related to our approach is Kernel Mean Matching (KMM) [11]. KMM is based on the observation that $E_q(\Phi(x)) = E_p(\frac{q}{p}\Phi(x))$, where $\Phi$ is the feature map corresponding to an RKHS $\mathcal{H}$. It is rewritten as an optimization problem

$$\frac{q(x)}{p(x)} = \arg \min_{\beta \in L_2, \beta(x) > 0, E_p(\beta) = 1} \|E_q(\Phi(x)) - E_p(\beta(x)\Phi(x))\|_{\mathcal{H}} \tag{5}$$

The quantity on the right can be estimated given a sample from $p$ and a sample from $q$ and the minimization becomes a quadratic optimization problem over the values of $\beta$ at the points sampled from $p$. Writing down the feature map explicitly, i.e., recalling that $\Phi(x) = K_{\mathcal{H}}(x, \cdot)$, we see that the equality $E_q(\Phi(x)) = E_p(\frac{q}{p}\Phi(x))$ is equivalent to the integral equation Eq. 2 considered as an identity in the Hilbert space $\mathcal{H}$. Thus the problem of KMM can be viewed within our setting Type I (see the Remark 2 in the introduction), with a RKHS norm but a different optimization algorithm.

However, while the KMM optimization problem in Eq. 5 uses the RKHS norm, the weight function $\beta$ itself is not in the RKHS. Thus, unlike most other algorithms in the RKHS framework (in particular, FIRE), the empirical optimization problem resulting from Eq. 5 does not have a natural out-of-sample extension[4].

Also, since there is no regularizing term, the problem is less stable (see Section 6 for some experimental comparisons) and the theoretical analysis is harder (however, see [9] and the recent paper [37] for some nice theoretical analysis of KMM in certain settings).

Another related recent algorithm is Least Squares Importance Sampling (LSIF) [14], which attempts to estimate the density ratio by choosing a parametric linear family of functions and choosing a function from this family to minimize the $L_{2,p}$ distance to the density ratio. A similar setting with the Kullback-Leibler distance (KLIEP) was proposed in [31]. This has an advantage of a natural out-of-sample extension property. We note that our method for unsupervised parameter selection in Section 6 is related to their ideas. However, in our case the set of test functions does not need to form a good basis since no approximation is required.

We note that our methods are closely related to a large body of work on kernel methods in machine learning and statistical estimation (e.g., [28, 24, 23]). Many of these algorithms can be interpreted as inverse problems, e.g., [6, 27] in the Tikhonov regularization or other regularization frameworks. In particular, we note interesting methods for density estimation proposed in [18] and estimating the support of density through spectral regularization in [7], as well as robust density estimation using RKHS formulations [15] and conditional density [10].

We also note the connections of the methods in this paper to properties of density-dependent operators in classification and clustering [35, 25]. There are also connections to geometry and density-dependent norms for semi-supervised learning, e.g., [3].

Finally, the setting in this paper is connected to the large literature on integral equations [16]. In particular, we note [34], which analyzes the classical Fredholm problem using regularization for noisy data.

## 3   Settings and Algorithms

### 3.1   Some preliminaries

We start by introducing some objects and function spaces important for our development. As usual, the space of square-integrable functions with respect to a measure $\rho$, is defined as follows:

$$L_{2,\rho} = \left\{ f : \int_\Omega |f(x)|^2 d\rho < \infty \right\}.$$

This is a Hilbert space with the inner product defined in the usual way by $\langle f, g \rangle_{2,\rho} = \int_\Omega f(x)g(x)d\rho$.

Given a function of two variables $k(x,y)$ (a kernel), we define the operator $\mathcal{K}_\rho$:

$$\mathcal{K}_\rho f(y) := \int_\Omega k(x,y)f(x)d\rho(x).$$

We will use the notation $\mathcal{K}_{t,\rho}$ to explicitly refer to the parameter of the kernel function $k_t(x,y)$, when it is a $\delta$-family.

If the function $k(x,y)$ is symmetric and positive definite, then there is a corresponding *Reproducing Kernel Hilbert space* (RKHS) $\mathcal{H}$. We recall the key property of the kernel $k_\mathcal{H}$: for any $f \in \mathcal{H}$, $\langle f, k_\mathcal{H}(x,\cdot)\rangle_\mathcal{H} = f(x)$. The direct consequence of this is the Representer Theorem, which allows us to write solutions to various optimization problems over $\mathcal{H}$ in terms of linear combinations of kernels supported on sample points (see [28] for an in-depth discussion or the RKHS theory and the issues related to learning).

It is important to note that in some of our algorithms the RKHS kernel $k_\mathcal{H}$ will be different from the kernel of the integral operator $k$.

Given a sample $x_1, \ldots, x_n$ from $p$, one can approximate the $L_{2,p}$ norm of a function[5] $f$ by $\|f\|_{2,p}^2 \approx \frac{1}{n}\sum_i |f(x_i)|^2$. Similarly, the integral operator $K_p f(x) \approx \frac{1}{n}\sum_i k(x_i,x)f(x_i)$. These approximate equalities can be made precise by using appropriate concentration inequalities.

### 3.2   The FIRE Algorithms

As discussed in the introduction, the starting point for our development is the integral equality

$$[\text{Type I}]: \quad \mathcal{K}_p \frac{q}{p}(x) = \int_\Omega k(x,y)\frac{q(y)}{p(y)}p(y)dy = \mathcal{K}_q \mathbf{1}(x). \tag{6}$$

Notice that in Type I, the kernel is not necessary to be in $\delta$-family. For example, it could be linear kernel. Thus, we omit the $t$ in the kernel for the Type I case.

Moreover, if the kernel $k_t(x,y)$ is a Gaussian, which we will analyze in detail, or another $\delta$-family and for $f$ sufficiently smooth $\mathcal{K}_{t,q}f(x) \approx f(x)p(x) + o(1)$ and hence

$$[\text{Type II}]: \quad \mathcal{K}_{t,p}\frac{q}{p}(x) = \int_\Omega k_t(x,y)\frac{q(y)}{p(y)}p(y)dy = q(x) + o(1). \tag{7}$$

In fact, for the Gaussian kernel, the $o(1)$ term is of the order $t$. Since it is important that the kernel $k_t$ is in the $\delta$-family with bandwidth $t$, so we keep $t$ in the notation in this case.

Assuming that either $\mathcal{K}_q\mathbf{1}$ or $q$ are known (for simplicity we will refer to these settings as Type I and Type II, respectively) these Eqs. 6,7 become integral equations for $\frac{p}{q}$, known as the Fredholm equations of the first kind.

To address the problem of estimating $\frac{p}{q}$ we need to obtain an approximation to the solution which (a) can be obtained computationally from sampled data, (b) is stable with respect to sampling and other perturbation of the input function[6] and, preferably, (c) can be analyzed using the standard machinery of functional analysis.

To provide a framework for solving these inverse problems we apply the classical techniques of regularization combined with the RKHS norm popular in machine learning. In particular a simple formulation of Eq.6 in terms of Tikhonov regularization with the $L_{2,p}$ norm is as follows:

$$[\text{Type I}]: \quad f_\lambda^{\text{I}} = \arg\min_{f \in \mathcal{H}} \|\mathcal{K}_p f - \mathcal{K}_q \mathbf{1}\|_{2,p}^2 + \lambda \|f\|_{\mathcal{H}}^2 \tag{8}$$

Here $\mathcal{H}$ is an appropriate Reproducing Kernel Hilbert Space. Similarly Eq. 7 can be written as

$$[\text{Type II}]: \quad f_\lambda^{\text{II}} = \arg\min_{f \in \mathcal{H}} \|\mathcal{K}_{t,p} f - q\|_{2,p}^2 + \lambda \|f\|_{\mathcal{H}}^2 \tag{9}$$

We will now discuss the empirical versions of these equations and the resulting algorithms in different settings and for different norms.

### 3.3 Algorithms for the Type I setting.

Given an iid sample from $p$, $\boldsymbol{z}_p = \{x_1, x_2, \ldots, x_n\}$ and an iid sample from $q$, $\boldsymbol{z}_q = \{x_1', x_2', \ldots, x_m'\}$ (we will denote the combined sample by $\boldsymbol{z}$) we can approximate the integral operators $\mathcal{K}_p$ and $\mathcal{K}_q$ by

$$K_{\boldsymbol{z}_p} f(x) = \frac{1}{n} \sum_{x_i \in \boldsymbol{z}_p} k(x_i, x) f(x_i) \quad \text{and} \quad K_{\boldsymbol{z}_q} f(x) = \frac{1}{m} \sum_{x_i' \in \boldsymbol{z}_q} k(x_i', x) f(x_i'). \tag{10}$$

Thus the empirical version of Eq. 8 becomes

$$f_{\lambda,\boldsymbol{z}}^{\text{I}} = \arg\min_{f \in \mathcal{H}} \frac{1}{n} \sum_{x_i \in \boldsymbol{z}_p} ((\mathcal{K}_{\boldsymbol{z}_p} f)(x_i) - (\mathcal{K}_{\boldsymbol{z}_q} \mathbf{1})(x_i))^2 + \lambda \|f\|_{\mathcal{H}}^2 \tag{11}$$

We observe that the first term of the optimization problem involves only evaluations of the function $f$ at the points of the sample $z_p$.

Thus, using the Representer Theorem and the standard matrix algebra manipulation we obtain the following solution:

$$f_{\lambda,\boldsymbol{z}}^{\text{I}}(x) = \sum_{x_i \in \boldsymbol{z}_p} k_{\mathcal{H}}(x_i, x) v_i \text{ and } \boldsymbol{v} = \left(K_{p,p}^2 K_{\mathcal{H}} + n\lambda I\right)^{-1} K_{p,p} K_{p,q} \mathbf{1}_{\boldsymbol{z}_q}. \tag{12}$$

where the kernel matrices are defined as follows: $(K_{p,p})_{ij} = \frac{1}{n} k(x_i, x_j)$, $(K_{\mathcal{H}})_{ij} = k_{\mathcal{H}}(x_i, x_j)$ for $x_i, x_j \in \boldsymbol{z}_p$ and $K_{p,q}$ is defined as $(K_{p,q})_{ij} = \frac{1}{m} k(x_i, x_j')$ for $x_i \in \boldsymbol{z}_p$ and $x_j' \in \boldsymbol{z}_q$.

To compute the whole regularization path for all $\lambda$'s, or computing the inverse for every $\lambda$, we can use the following formula for $\boldsymbol{v}$:

$$\boldsymbol{v} = Q(\Lambda + n\lambda I)^{-1} Q^{-1} K_{p,p} K_{p,q} \mathbf{1}_{\boldsymbol{z}_q},$$

where $K_{p,p}^2 K_{\mathcal{H}} = Q\Lambda Q^{-1}$ is a diagonalization[7] of $K_{p,p}^2 K_{\mathcal{H}}$ (i.e., $\Lambda$ is diagonal).

When $K_{\mathcal{H}}$ and $K_{p,p}$ are obtained using the same kernel function $k$, i.e. $\frac{1}{n} K_{\mathcal{H}} = K_{p,p}$, the expression simplifies:

$$\boldsymbol{v} = \frac{1}{n} \left(K_{p,p}^3 + \lambda I\right)^{-1} K_{p,p} K_{p,q} \mathbf{1}_{\boldsymbol{z}_q}.$$

In that case (or, more generally, if they commute) the diagonalization is obtained by computing the eigen-decomposition of $K_{p,p} = Q\Lambda Q^T$, where $Q$ is an orthogonal matrix. Then the solution could be computed using the following formula:

$$f_{\lambda,\boldsymbol{z}}^{\text{I}}(x) = \frac{1}{n} \sum_{x_i \in \boldsymbol{z}_p} k(x_i, x) v_i \text{ and } \boldsymbol{v} = Q \left(\Lambda^3 + \lambda I\right)^{-1} \Lambda Q^T K_{p,q} \mathbf{1}_{\boldsymbol{z}_q}.$$

Similarly to many other algorithms based on the square loss function, this formulation allows us to efficiently compute the solution for many values of the parameter $\lambda$ simultaneously, which is very useful for cross-validation.

### 3.3.1 Algorithms for $\gamma L_{2,p} + (1-\gamma)L_{2,q}$ norm.

Depending on the setting, we may want to minimize the error of the estimate over the probability distribution $p$, $q$ or over some linear combination of these. A significant potential benefit of using a linear combination is that both samples can be used at the same time in the loss function. First we state the continuous version of the problem:

$$f_\lambda^* = \arg\min_{f \in \mathcal{H}} \; \gamma \|\mathcal{K}_p f - \mathcal{K}_q \mathbf{1}\|_{2,p}^2 + (1-\gamma)\|\mathcal{K}_p f - \mathcal{K}_q \mathbf{1}\|_{2,q}^2 + \lambda \|f\|_{\mathcal{H}}^2 \tag{13}$$

Given a sample from $p$, $\boldsymbol{z}_p = \{x_1, x_2, \ldots, x_n\}$ and a sample from $q$, $\boldsymbol{z}_q = \{x_1', x_2', \ldots, x_m'\}$ we obtain an empirical version of the Eq. 13:

$$f_{\lambda,\boldsymbol{z}}^*(x) = \arg\min_{f \in \mathcal{H}} \frac{\gamma}{n} \sum_{x_i \in \boldsymbol{z}_p} \left((\mathcal{K}_{\boldsymbol{z}_p} f)(x_i) - (\mathcal{K}_{\boldsymbol{z}_q}\mathbf{1})(x_i)\right)^2 + \frac{1-\gamma}{m}\sum_{x_i' \in \boldsymbol{z}_q}\left((\mathcal{K}_{\boldsymbol{z}_p}f)(x_i') - (\mathcal{K}_{\boldsymbol{z}_q}\mathbf{1})(x_i')\right)^2 + \lambda\|f\|_H^2$$

Using the Represeter Theorem we can derive:

$$f_{\lambda,\boldsymbol{z}}^*(x) = \sum_{x_i \in \boldsymbol{z}_p} v_i k_{\mathcal{H}}(x_i, x) \qquad \boldsymbol{v} = (K + n\lambda I)^{-1} K_1 \mathbf{1}_{\boldsymbol{z}_q}$$

where

$$K = \left(\frac{\gamma}{n}(K_{p,p})^2 + \frac{1-\gamma}{m}K_{q,p}^T K_{q,p}\right)K_{\mathcal{H}} \;\; \text{and} \;\; K_1 = \left(\frac{\gamma}{n}K_{p,p}K_{p,q} + \frac{1-\gamma}{m}K_{q,p}^T K_{q,q}\right)$$

Here $(K_{p,p})_{ij} = \frac{1}{n}k(x_i, x_j)$, $(K_{\mathcal{H}})_{ij} = k_{\mathcal{H}}(x_i, x_j)$ for $x_i, x_j \in \boldsymbol{z}_p$. $K_{p,q}$ and $K_{q,p}$ are defined as $(K_{p,q})_{ij} = \frac{1}{m}k(x_i, x_j')$ and $(K_{q,p})_{ji} = \frac{1}{n}k(x_j', x_i)$ for $x_i \in \boldsymbol{z}_p, x_j' \in \boldsymbol{z}_q$.

We see that despite the loss function combining both samples, the solution is still a summation of kernels over the points in the sample from $p$.

### 3.3.2 Algorithms for the RKHS norm.

In addition to using the RKHS norm for regularization norm, we can also use it as a loss function:

$$f_\lambda^* = \arg\min_{f \in \mathcal{H}} \|\mathcal{K}_p f - \mathcal{K}_q \mathbf{1}\|_{\mathcal{H}'}^2 + \lambda\|f\|_{\mathcal{H}}^2 \tag{14}$$

Here the Hilbert space $\mathcal{H}'$ must correspond to the convolution kernel $k$ and can potentially be different from the space $\mathcal{H}$ used for regularization. Note that this formulation is only applicable in the Type I setting since it requires the function $q$ to belong to the RKHS $\mathcal{H}'$.

Given two samples $\boldsymbol{z}_p, \boldsymbol{z}_q$, it is straightforward to write down the empirical version of this problem, leading to the following formula:

$$f_{\lambda,\boldsymbol{z}}^*(x) = \sum_{x_i \in \boldsymbol{z}_p} v_i k_{\mathcal{H}}(x_i, x) \qquad \boldsymbol{v} = (K_{p,p}K_{\mathcal{H}} + n\lambda I)^{-1} K_{p,q}\mathbf{1}_{\boldsymbol{z}_q}. \tag{15}$$

The result is somewhat similar to our Type I formulation with the $L_{2,p}$ norm. We note the connection between this formulation of using the RKHS norm as a loss function and the KMM algorithm [11]. The Eq. 15 can be viewed as a regularized version of KMM (with a different optimization procedure), when the kernels $K$ and $K_{\mathcal{H}}$ are the same.

Interestingly a somewhat similar formula arises in [14] as unconstrained LSIF, with a different functional basis (kernels centered at the points of the sample $\boldsymbol{z}_q$) and in a setting not directly related to RKHS inference.

## 3.4 Algorithms for the Type II and 1.5 settings.

In the Type II setting we assume that we have a sample $\boldsymbol{z} = \{x_1, x_2, \ldots, x_n\}$ drawn from $p$ and that we know the function values $q(x_i)$ at the points of the sample.

Replacing the norm and the integral operator with their empirical versions, we obtain the following optimization problem:

$$f_{\lambda,\boldsymbol{z}}^{II} = \arg\min_{f \in \mathcal{H}} \frac{1}{n}\sum_{x_i \in \boldsymbol{z}} (\mathcal{K}_{t,\boldsymbol{z}} f(x_i) - q(x_i))^2 + \lambda\|f\|_{\mathcal{H}}^2 \tag{16}$$

Recall that $\mathcal{K}_{t,\boldsymbol{z}}$ is the empirical version of $\mathcal{K}_{t,p}$ defined by

$$\mathcal{K}_{t,\boldsymbol{z}} f(x) = \frac{1}{n} \sum_{x_i \in \boldsymbol{z}} k_t(x_i, x) f(x_i)$$

As before, using the Representer Theorem we obtain an analytical formula for the solution:

$$f_{\lambda,\boldsymbol{z}}^{\mathrm{II}}(x) = \sum_{x_i \in \boldsymbol{z}} k_{\mathcal{H}}(x_i, x) v_i \text{ where } \boldsymbol{v} = \left(K^2 K_{\mathcal{H}} + n\lambda I\right)^{-1} K\mathbf{q}. \qquad (17)$$

where the kernel matrix $K$ is defined by $K_{ij} = \frac{1}{n} k_t(x_i, x_j)$, $(K_{\mathcal{H}})_{ij} = k_{\mathcal{H}}(x_i, x_j)$ and $\mathbf{q}_i = q(x_i)$.

### 3.4.1   Type 1.5: The setting and the algorithm.

This case (see Eq. 4) is intermediate between Type I and Type II. The setting is the same as in Type I, in that we are given two samples $z_p$ from $p$ and $z_q$ from $q$. But similarly to Type II, we use the fact that $\mathcal{K}_p \frac{q}{p} \approx \mathcal{K}_q' \mathbf{1}$ when $\mathcal{K}_p$ and $\mathcal{K}_q'$ are different $\delta$-function-like kernels (e.g., two Gaussians of different bandwidth). The algorithm is similar to that for Type I with the difference that the kernel matrix $K_{q,q}'$ is computed using the kernel $k'(x, y)$: $(K_{q,q}')_{ij} = \frac{1}{m} k'(x_i, x_j')$.

$$f_{\lambda,\boldsymbol{z}}^{1.5}(x) = \sum_{x_i \in \boldsymbol{z}_p} k_{\mathcal{H}}(x_i, x) v_i \text{ and } \boldsymbol{v} = \left(K_{p,p}^2 K_{\mathcal{H}} + n\lambda I\right)^{-1} K_{p,p} K_{q,q}' \mathbf{1}_{\boldsymbol{z}_q}.$$

### 3.5   Spectral Cutoff Regularization

In this section we briefly discuss an alternative form of regularization, based on thresholding the spectrum of the kernel matrix. It also leads to simple algorithms comparable to those for Tikhonov regularization and may have certain computational advantages.

Since $\mathcal{K}_p$ is a compact self-adjoint operator on $L_{2,p}$, its eigenfunctions $\{u_0, u_1, \dots\}$ form a complete orthogonal basis for $L_{2,p}$. An alternative method of regularization is the so-called *spectral cutoff* where the problem is restricted to the subspace spanned by the top few eigenfunctions of $\mathcal{K}_p$ Thus the regularization problems become

$$f_{\lambda}^{\mathrm{I,spec}} = \arg\min_{f \in \mathcal{H}_k} \|\mathcal{K}_p f - \mathcal{K}_q \mathbf{1}\|_{2,p}^2$$

$$f_{\lambda}^{\mathrm{II,spec}} = \arg\min_{f \in \mathcal{H}_{t,k}} \|\mathcal{K}_{t,p} f - q\|_{2,p}^2$$

where $\mathcal{H}_k$ and $\mathcal{H}_{t,k}$ is the finite dimensional subspace of $L_{2,p}$ spanned by the eigenvectors of $\mathcal{K}_p$ and $\mathcal{K}_{t,p}$ corresponding to the $k$ largest eigenvalues.

Without going into detail, it can be seen that the corresponding empirical optimization problems are

$$f_{\lambda,\boldsymbol{z}}^{\mathrm{I,spec}} = \arg\min_{f \in H_{k,\boldsymbol{z}}} \frac{1}{n} \sum_{x_i \in \boldsymbol{z}_p} (\mathcal{K}_{\boldsymbol{z}_p} f(x_i) - \mathcal{K}_{t,\boldsymbol{z}_q} \mathbf{1}(x_i))^2 \qquad (18)$$

$$f_{\lambda,\boldsymbol{z}}^{\mathrm{II,spec}} = \arg\min_{f \in H_{t,k,\boldsymbol{z}}} \frac{1}{n} \sum_{x_i \in \boldsymbol{z}} (\mathcal{K}_{t,\boldsymbol{z}_p} f(x_i) - q(x_i))^2 \qquad (19)$$

where the span of eigenvectors of the kernel matrix $K$ is taken instead of the eigenfunctions of $\mathcal{K}_p$ or $\mathcal{K}_{t,p}$.

For this algorithm, we assume $K_{\mathcal{H}}$ and $K_1$ use the same kernel. Then the solution to the empirical regularization problems given in Eqs. 18,19 are respectively

$$f_{\lambda,\boldsymbol{z}}^{\mathrm{I,spec}}(x) = \frac{1}{n} \sum_{x_i \in \boldsymbol{z}_p} k(x_i, x) v_i$$

$$\boldsymbol{v} = Q_k \Lambda_k^{-2} Q_k^T K_2 \mathbf{1}_{\boldsymbol{z}_q} \qquad (20)$$

$$f_{\lambda,\boldsymbol{z}}^{\text{II,spec}}(x) = \frac{1}{n} \sum_{x_i \in \boldsymbol{z}} k_t(x_i, x) v_i$$

$$\boldsymbol{v} = Q_k \Lambda_k^{-2} Q_k^T q$$

(21)

where $K_1 = Q \Lambda Q^T$ is the eigendecomposition of $K_1$ with orthogonal matrix $Q$ and diagonal matrix $\Lambda$, and $Q_k$ and $\Lambda_k$ is the submatrices of $Q$ and $\Lambda$ corresponding to the $k$ largest eigenvalues of the kernel matrix $K_1$ and the remaining objects are defined in the previous subsection.

We note that spectral regularization can be faster computationally as it requires to compute only the top few eigenvectors of the kernel matrix. There are several efficient algorithms for computing eigen-decomposition when only the first $k$ eigenvalues are needed. Thus spectral regularization can be more computationally efficient than the Tikhonov regularization which potentially requires a full eigen-decomposition or matrix multiplication.

### 3.6 Comparison of type I and type II settings.

While at first glance the type II, setting may appear to be more restrictive than type I, there are a number of important differences in their applicability.

1. In Type II setting $q$ does not have to be a density function (i.e., non-negative and integrate to one).

2. Eq. 11 of the Type I setting cannot be easily solved in the absence of a sample $\boldsymbol{z}_q$ from $q$, since estimating $\mathcal{K}_q$ requires either sampling from $q$ (if it is a density) or estimating the integral in some other way, which may be difficult in high dimension but perhaps of interest in certain low-dimensional application domains.

3. There are a number of problems (e.g., many problems involving MCMC) where $q(x)$ is known explicitly (possibly up to a multiplicative constant), while sampling from $q$ is expensive or even impossible computationally [19].

4. Unlike Eq. 8, Eq. 9 has an error term depending on the kernel, which is essentially the difference between the kernel and the $\delta$-function. For example, in the important case of the Gaussian kernel, the error is of the order $O(t)$, where $t$ is the variance.

5. While a number of different norms are available in the Type I setting, only the $L_{2,p}$ norm is available for Type II.

## 4 Theoretical analysis: bounds and convergence rates for Gaussian Kernels

In this section, we state our main results on bounds and convergence rates for our algorithm based on Tikhonov regularization with a Gaussian kernel. We consider both Type I and Type II settings for the Euclidean and manifold cases and make a remark on the Euclidean domains with boundary.

To simplify the theoretical development the integral operator and the RKHS $\mathcal{H}$ will correspond to the same Gaussian kernel $k_t(x, y)$. Most of the proofs will be given in the next Section 5. We note that two Gaussian kernels with different bandwidth parameters can be analyzed using only minor modifications to our arguments.

### 4.1 Assumptions

Before proceeding to the main results, we will state the assumptions on the density functions $p$ and $q$ and the basic setting for our theorems:

1. The set $\Omega$ where the density function $p$ is defined could be one of the following: (1) the whole $\mathbb{R}^d$; (2) a compact smooth Riemannian sub-manifold $\mathcal{M}$ of $d$-dimension in $\mathbb{R}^n$. In both cases, we need $p(x) < \Gamma$ and $q(x) < \Gamma$ for any $x \in \Omega$. We will also make some remarks about a compact domain in $\mathbb{R}^d$ with boundary.

2. We also require $\frac{q}{p}, \frac{q}{p^2} \in W_2^2(\Omega)$, where $W_2^2(\Omega)$ is the Sobolev space of functions on $\Omega$ (e.g., [32]). The properties of $W_2^2(\Omega)$ we need will be discussed later in the proof.

It will be important for us that $\mathcal{H}$ is isometric to $L_{2,p}$ under the map $\mathcal{K}_p^{1/2} : L_{2,p} \to \mathcal{H}$, that is, $\|f\|_{\mathcal{H}} = \|\mathcal{K}_p^{-1/2} f\|_{L_{2,p}}$ for any $f \in \mathcal{H}$. Here the integral operator $\mathcal{K}_p$ uses the RKHS kernel corresponding to $\mathcal{H}$.

## 4.2 Main Theorems

### 4.2.1 Type I setting

We will provide theoretical results for our setting Type I, where both the operator and the regularization kernel are Gaussian $k_t(x,y) = \frac{1}{(2\pi t)^{d/2}} e^{-\frac{\|x-y\|^2}{2t}}$ with the same bandwidth parameter $t$.

**Theorem 1.** *Let $p$ and $q$ be two density functions on $\Omega$ and $q$ be another density over $\Omega$ satisfying the assumption in Sec. 4.1. Given $n$ points, $\boldsymbol{z}_p = \{x_1, x_2, \ldots, x_n\}$, i.i.d. sampled from $p$ and $m$ points, $\boldsymbol{z}_q = \{x'_1, x'_2, \ldots, x'_m\}$, i.i.d. sampled from $q$, and for small enough $t$, for the solution to the optimization problem in (11), with confidence at least $1 - 2e^{-\tau}$, we have*

*(1) If the domain $\Omega$ is $\mathbb{R}^d$,*

$$
\begin{aligned}
\left\| f_{\lambda,\boldsymbol{z}}^I - \frac{q}{p} \right\|_{2,p} \leq & C_1 t + C_2 \lambda^{\frac{1}{2}} \quad \text{(Approximating Error)} \\
& + C_3 \frac{\sqrt{\tau}}{\lambda t^{d/2}} \left( \frac{1}{\sqrt{m}} + \frac{1}{\lambda^{1/6}\sqrt{n}} \right) \quad \text{(Sampling Error)},
\end{aligned}
\tag{22}
$$

*where $C_1, C_2, C_3$ are constants independent of $t, \lambda$.*

*(2) If the domain $\Omega$ is a compact manifold without boundary of $d$ dimension, for any $1 < \varepsilon < 1$,*

$$
\begin{aligned}
\left\| f_{\lambda,\boldsymbol{z}}^I - \frac{q}{p} \right\|_{2,p} \leq & C_1 t^{1-\varepsilon} + C_2 \lambda^{\frac{1}{2}} \quad \text{(Approximating Error)} \\
& + C_3 \frac{\sqrt{\tau}}{\lambda t^{d/2}} \left( \frac{1}{\sqrt{m}} + \frac{1}{\lambda^{1/6}\sqrt{n}} \right) \quad \text{(Sampling Error)},
\end{aligned}
\tag{23}
$$

*where $C_1, C_2, C_3$ are constants independent of $t, \lambda$.*

*Proof.* See Section 5. $\qquad\qquad \square$

As a consequence we obtain the following corollary establishing the convergence rates:

**Corollary 2.** *Assuming $m > \lambda^{1/3}n$, with confidence at least $1 - 2e^{-\tau}$, we have the following:*

*(1) If $\Omega = \mathbb{R}^d$,*

$$
\left\| f_{\lambda,\boldsymbol{z}}^I - \frac{q}{p} \right\|_{2,p}^2 = O\left( \sqrt{\tau} n^{-\frac{1}{3.5s+d/2}} \right)
$$

*(2) If $\Omega$ is a $d$-dimensional sub-manifold of a Euclidean space,*

$$
\left\| f_{\lambda,\boldsymbol{z}}^I - \frac{q}{p} \right\|_{2,p}^2 = O\left( \sqrt{\tau} n^{-\frac{1}{3.5+d/2}} \right)
$$

*Proof.* For the Euclidean space, set $t = n^{-\frac{1}{\frac{10.5}{3}s+d}}, \lambda = n^{-\frac{s}{\frac{10.5}{3}s+d}}$ and apply Theorem 1 (Eq. 22 for the Euclidean case). For the sub-manifold case set $t = n^{-\frac{1}{7+d}}, \lambda = n^{-\frac{2}{7+d}}$. $\qquad \square$

### 4.2.2 Type II setting

In this section we provide an analysis for the Type II setting and also make a remark about the error analysis for the compact domains in $\mathbb{R}^d$.

Recall that in Type II setting we have a set of points sampled from $p$ and assume that the values of $q$ on those points are known. Note, that $q$ does not have to be a density function.

**Theorem 3.** *Let $p$ be a density function on $\Omega$ and $q$ be a function satisfying the assumptions in Sec. 4.1. Given $n$ points $\boldsymbol{z} = \{x_1, x_2, \ldots, x_n\}$ sampled i.i.d. from $p$, and for sufficiently small $t$, for the solution to the optimization problem in (16), with confidence at least $1 - 2e^{-\tau}$, we have*

*(1) If the domain $\Omega$ is $\mathbb{R}^d$,*

$$\left\| f_{\lambda,\boldsymbol{z}}^{II} - \frac{q}{p} \right\|_{2,p} \leq C_1 t + C_2 \lambda^{\frac{1}{2}} + C_3 \lambda^{-\frac{1}{3}} \left\| \mathcal{K}_{t,q} \mathbf{1} - q \right\|_{2,p} + C_4 \frac{\sqrt{\tau}}{\lambda^{3/2} t^{d/2} \sqrt{n}}, \qquad (24)$$

*where $C_1, C_2, C_3, C_4$ are constants independent of $t, \lambda$. Moreover, $\|\mathcal{K}_{t,q}\mathbf{1} - q\|_{2,p} = O(t)$.*

*(2) If $\Omega$ is a $d$-dimensional sub-manifold of a Euclidean space, for any $1 < \varepsilon < 1$,*

$$\left\| f_{\lambda,\boldsymbol{z}}^{II} - \frac{q}{p} \right\|_{2,p} \leq C_1 t^{1-\varepsilon} + C_2 \lambda^{1/2} + C_3 \lambda^{-\frac{1}{3}} \left\| \mathcal{K}_{t,q} \mathbf{1} - q \right\|_{2,p} + C_4 \frac{\sqrt{\tau}}{\lambda^{3/2} t^{d/2} \sqrt{n}}, \qquad (25)$$

*where $C_1, C_2, C_3, C_4$ are constants independent of $t, \lambda$. Moreover, $\|\mathcal{K}_{t,q}\mathbf{1} - q\|_{2,p} = O(t^{1-\varepsilon})$ for any $\varepsilon > 0$.*

**Remark.** It can be shown that if $\Omega$ is a compact subset with sufficiently smooth boundary in $\mathbb{R}^d$, we have the same bound with (1) except for $\|\mathcal{K}_{t,q}\mathbf{1} - q\|_{2,p} = O(t^{\frac{1}{4}-\varepsilon})$ for any any $\varepsilon > 0$.

As before, we obtain the rates as a corollary:

**Corollary 4.** *With confidence at least $1 - 2e^{-\tau}$, we have:*

*(1) If $\Omega = \mathbb{R}^d$,*

$$\left\| f_{\lambda,\boldsymbol{z}}^{II} - \frac{q}{p} \right\|_{2,p}^2 = O\left( \sqrt{\tau} n^{-\frac{1}{4+\frac{5}{6}d}} \right)$$

*(2) If $\Omega$ is a $d$-dimensional sub-manifold of a Euclidean space, than for any $0 < \varepsilon < 1$*

$$\left\| f_{\lambda,\boldsymbol{z}}^{II} - \frac{q}{p} \right\|_{2,p}^2 = O\left( \sqrt{\tau} n^{-\frac{1-\varepsilon}{4-4\varepsilon+\frac{5}{6}d}} \right)$$

*Proof.* For the case of $\mathbb{R}^d$, set $t = n^{-\frac{1}{4.8+d}}, \lambda = n^{-\frac{1}{4+\frac{5}{6}d}}$. For case of sub-manifold case, set $t = n^{-\frac{1-\varepsilon}{4.8-4.8\varepsilon+d}}, \lambda = n^{-\frac{1-\varepsilon}{4-4\varepsilon+\frac{5}{6}d}}$. Apply Theorem 3. $\qquad \square$

# 5 Proofs of Theorems

In this section, we provide a proof for the our main Theorem 1 for setting I. The proof for the Theorem 3 for the setting type II is along similar lines and can be found in the appendix.

## 5.1 Basics about RKHS

For any probability measure $\rho$, $\mathcal{K}_{t,\rho}$ is a self-adjoint operator. Thus its eigenfunctions $\{u_{0,t}, u_{1,t}, \ldots\}$ form a complete orthogonal basis for $L_{2,\rho}$. Denote the eigenvalues of $\mathcal{K}_{t,\rho}$ by $\{\sigma_{0,t}, \sigma_{1,t}, \ldots\}$. The norm of $\mathcal{K}_{t,\rho}$, $\|\mathcal{K}_{t,\rho}\|_{L_{2,\rho} \to L_{2,\rho}} \leq \max_i \sigma_{i,t} < c$ for a constant $c$. We know that $H_t$ is isometric to $L_{2,\rho}$ under the map $\mathcal{K}_{t,\rho}^{1/2} : L_{2,\rho} \to H_t$, i.e. $\|f\|_{H_t} = \|\mathcal{K}_{t,\rho}^{-1/2} f\|_{L_{2,\rho}}$ for any $f \in H_t$, and this is the definition we use for the norm $\|\cdot\|_{H_t}$ of $H_t$. This also implies that $\|\mathcal{K}_{t,\rho}^{-1/2} f\|_{L_{2,\rho}} < \infty$ for any $f \in H_t$. And $\mathcal{K}_{t,\rho}$ is defined using the spectrum of $\mathcal{K}_{t,\rho}$,

$$\mathcal{K}_{t,\rho} f = \sum_i \sigma_{i,\rho} \langle f, u_{i,t} \rangle u_{i,t}$$

## 5.2 Proof of Theorem 1

*Proof.* Recall the definition of $f_\lambda^{\mathrm{I}}$ and $f_{\lambda,\boldsymbol{z}}^{\mathrm{I}}$ in Eq. 8 and Eq. 11. By the triangle inequality, we have

$$\left\| \frac{q}{p} - f_{\lambda,\boldsymbol{z}}^{\mathrm{I}} \right\|_{2,p} \leq \left\| \frac{q}{p} - f_\lambda^{\mathrm{I}} \right\|_{2,p} + \left\| f_\lambda^{\mathrm{I}} - f_{\lambda,\boldsymbol{z}}^{\mathrm{I}} \right\|_{2,p}. \tag{26}$$
$$= (\text{Approximation Error}) + (\text{Sampling Error})$$

The *approximation error* $\left\| f_\lambda^{\mathrm{I}} - \frac{q}{p} \right\|_{2,p}$ is a measure of the distance between $\frac{q}{p}$ and the optimal approximation given by algorithm (8) given infinite number of data. The *sampling error* term $\left\| f_\lambda^{\mathrm{I}} - f_{\lambda,\boldsymbol{z}}^{\mathrm{I}} \right\|_{2,p}$ the difference between $f_\lambda^{\mathrm{I}}$ and $f_{\lambda,\boldsymbol{z}}^{\mathrm{I}}$, depending on the data points.

As typical in these types of estimates our proof consists of two parts: bounding the approximating error, $\left\| f_\lambda^{\mathrm{I}} - \frac{q}{p} \right\|_{2,p}$ in Lemma 8 and providing a concentration bound for $\left\| f_{\lambda,\boldsymbol{z}}^{\mathrm{I}} - f_\lambda^{\mathrm{I}} \right\|_{2,p}$ in Lemma 9. The theorem follows immediately by putting these two results together. $\qquad\square$

### 5.2.1 Bound for Approximation Error

First of all, let us present three lemmas that are useful for bounding the approximation error.

**Lemma 5.** *Let $\lambda > 0$. If function $f \in W_s^2(\mathbb{R}^d)$ and $p(x) > 0$ for any $x \in \mathbb{R}^d$, then*

$$\arg \min_{g \in L_{2,p}} \left( \left\| f - \mathcal{K}_{t,p}^{1/2} g \right\|_{2,p}^2 + \lambda \|g\|_{2,p}^2 \right) \leq D_1 \|f\|_{W_s^2}^2 t^s + \lambda D_2 \|f\|_2^2. \tag{27}$$

*for constants $D_1, D_2$.*

*Proof.* See Appendix A. $\qquad\square$

**Lemma 6.** *Let $\lambda > 0$. If function $f \in W_2^2(\mathcal{M})$ defined on a compact Riemann sub-manifold of $d$-dimension in a Euclidean space, then*

$$\arg \min_{g \in L_{2,p}} \left( \left\| f - \mathcal{K}_{t,p}^{1/2} g \right\|_{2,p}^2 + \lambda \|g\|_{2,p}^2 \right) \leq D_1 \|f\|_{W_2^2}^2 t^2 + \lambda D_2 \|f\|_{2,p}^2. \tag{28}$$

*for constants $D_1, D_2$.*

*Proof.* See Appendix B $\qquad\square$

**Lemma 7.** *Suppose $p$ are a density function of probability measures of the domain $\Omega$ and satisfying the assumptions we gave in section 4.1. For any $f \in W_2^2(\Omega)$, we have the following: (1) When $\Omega$ is $\mathbb{R}^d$ and $f \in W_2^2(\mathbb{R}^2)$, we have*

$$\|\mathcal{K}_t f - f\|_{2,p} = O(t)$$

*(2) When $\Omega$ is a manifold $\mathcal{M}$ without boundary of $d$ dimension and $f \in W_2^2(\mathcal{M})$, we have*

$$\|\mathcal{K}_t f - f\|_{2,p} = O(t^{1-\varepsilon})$$

*for any $0 < \varepsilon < 1$.*

*Proof.* See Appendix C. $\qquad\square$

Now we can present the lemma that gives the bound of the approximation error in the following lemma.

**Lemma 8.** *Let $p, q$ be two density functions of probability measure over a domain $X$ satisfying the assumptions in 4.1. The solution to the optimization problem in (8), $f_\lambda^I$, satisfies the following inequality,*

*(1) when the domain $X$ is $\mathbb{R}^d$,*

$$\left\| f_\lambda^I - \frac{q}{p} \right\|_{2,p} \leq C_1 t + C_2 \lambda^{1/2}$$

*for constants $C_1, C_2$ which are independent of $\lambda$ and $t$.*

*(2) when the domain $X$ is a compact Riemannian sub-manifold $\mathcal{M}$ of $d$ dimension in $\mathbb{R}^N$,*

$$\left\| f_\lambda^I - \frac{q}{p} \right\|_{2,p} \leq C_1 t^{1-\varepsilon} + C_2 \lambda^{1/2}$$

*for any $1 < \varepsilon < 1$, and constants $C_1, C_2$ which are independent of $\lambda$ and $t$.*

*Proof.* Recall the equation (8). By functional calculus, we have analytical formula for $f_\lambda^I$ as follows,

$$f_\lambda^I = \sum_i \frac{\sigma_{i,t}^2}{\sigma_{i,t}^3 + \lambda} \langle \mathcal{K}_{t,q}\mathbf{1}, u_{i,t} \rangle_2 u_{i,t} = \left( \mathcal{K}_{t,p}^3 + \lambda \mathcal{I} \right)^{-1} \mathcal{K}_{t,p}^2 \mathcal{K}_{t,q}\mathbf{1} = \left( \mathcal{K}_{t,p}^3 + \lambda \mathcal{I} \right)^{-1} \mathcal{K}_{t,p}^3 \frac{q}{p}.$$

The last equation is because

$$\mathcal{K}_{t,q}\mathbf{1} = \mathcal{K}_{t,p}\frac{q}{p}.$$

Thus the approximating error is

$$\left\| f_\lambda^I - \frac{q}{p} \right\|_{2,p} = \left\| \left( \mathcal{K}_{t,p}^3 + \lambda \mathcal{I} \right)^{-1} \mathcal{K}_{t,p}^3 \frac{q}{p} - \frac{q}{p} \right\|_{2,p} \tag{29}$$

Notice that $\left( \mathcal{K}_{t,p}^3 + \lambda \mathcal{I} \right)^{-1} \mathcal{K}_{t,p}^3 \frac{q}{p}$ in (29) can also be rewritten as

$$\left( \mathcal{K}_{t,p}^3 + \lambda \mathcal{I} \right)^{-1} \mathcal{K}_{t,p}^3 \frac{q}{p} = \mathcal{K}_{t,p}^{3/2} g^*, \text{ and } g^* = \arg\min_{g \in L_{2,p}} \left\| \frac{q}{p} - \mathcal{K}_{t,p}^{3/2} g \right\|_{2,p}^2 + \lambda \|g\|_{2,p}^2$$

Thus,

$$\left\| \left( \mathcal{K}_{t,p}^3 + \lambda \mathcal{I} \right)^{-1} \mathcal{K}_{t,p}^3 \frac{q}{p} - \frac{q}{p} \right\|_{2,p}^2 \leq \min_{g \in L_{2,p}} \left\| \frac{q}{p} - \mathcal{K}_{t,p}^{3/2} g \right\|_{2,p}^2 + \lambda \|g\|_{2,p}^2 \tag{30}$$

The minimum of the above optimization problem can always be bounded by any specific $g \in L_{2,p}$. And we will expend the above formula such that we can take advantages of Lemma 5, 6 and 7. To this end, we define an map

$$\mathcal{T}(f, \lambda) = \arg\min_{g \in L_{2,p}} \left\| f - \mathcal{K}_{t,p}^{1/2} g \right\|_{2,p}^2 + \lambda \|g\|_{2,p}^2 := g^*$$

By functional calculus, it is not hard to see that $\mathcal{K}_{t,p}^{1/2} g^* = \left( \mathcal{K}_{t,p} + \lambda \mathcal{I} \right)^{-1} \mathcal{K}_{t,p} f$. Now let $\hat{g} = \mathcal{T}\left( \frac{q}{p^2}, \lambda \right)$. Now we could expend (30),

$$\min_{g \in L_{2,p}} \left\| \frac{q}{p} - \mathcal{K}_{t,p}^{3/2} g \right\|_{2,p}^2 + \lambda \|g\|_{2,p}^2 \leq \left\| \frac{q}{p} - \mathcal{K}_{t,p}^{3/2} \hat{g} \right\|_{2,p}^2 + \lambda \|\hat{g}\|_{2,p}^2$$

$$= \left\| \frac{q}{p} - \mathcal{K}_t \frac{q}{p} + \mathcal{K}_{t,p} \frac{q}{p^2} - \mathcal{K}_{t,p} \left( \mathcal{K}_{t,p}^{1/2} \hat{g} \right) \right\|_{2,p}^2 + \lambda \|\hat{g}\|_{2,p}^2$$

By inequality $(a+b)^2 \leq 2(a^2 + b^2)$, we have

$$\min_{g \in L_{2,p}} \left\| \frac{q}{p} - \mathcal{K}_{t,p}^{3/2} g \right\|_{2,p}^2 + \lambda \|g\|_{2,p}^2$$

$$\leq 2 \left\| \frac{q}{p} - \mathcal{K}_t \frac{q}{p} \right\|_{2,p}^2 + 2 \left( \left\| \mathcal{K}_{t,p} \left( \frac{q}{p^2} - \mathcal{K}_{t,p}^{1/2} \hat{g} \right) \right\|_{2,p}^2 + \lambda \|\hat{g}\|_{2,p}^2 \right)$$

$$\leq 2 \left\| \frac{q}{p} - \mathcal{K}_t \frac{q}{p} \right\|_{2,p}^2 + 2c \left( \left\| \frac{q}{p^2} - \mathcal{K}_{t,p}^{1/2} \hat{g} \right\|_{2,p}^2 + \frac{\lambda}{c} \|\hat{g}\|_{2,p}^2 \right)$$

The last inequality is because $\|\mathcal{K}_{t,p}\|_{L_{2,p}\to L_{2,p}} < c$ for constant $c > 1$. Note that we assume $\frac{q}{p}$ and $\frac{q}{p^2}$ are in $W_2^2(\Omega)$. Thus, the first term could be bounded using Lemma 7. For density over $\mathbb{R}^d$,

$$2\left\|\frac{q}{p} - \mathcal{K}_t\frac{q}{p}\right\|_{2,p} \le Lt,$$

and for manifold case, we have

$$2\left\|\frac{q}{p} - \mathcal{K}_t\frac{q}{p}\right\|_{2,p} \le Lt^{1-\varepsilon}.$$

for some constant $L > 0$.

Now we can apply Lemma 5 and 6 to get the bounds for the second term. By Lemma 5, for the densities $p, q$ over $\mathbb{R}^d$, we have

$$\left\|\frac{q}{p^2} - \mathcal{K}_{t,p}^{1/2}\hat{g}\right\|_{2,p}^2 + \frac{\lambda}{c}\|\hat{g}\|_{2,p}^2 = \min_{g\in L_{2,p}}\left\|\frac{q}{p^2} - \mathcal{K}_{t,p}^{1/2}g\right\|_{2,p}^2 + \frac{\lambda}{c}\|g\|_{2,p}^2 \le D_1\left\|\frac{q}{p^2}\right\|_{W_2^2}^2 t^2 + D_2\frac{\lambda}{c}\left\|\frac{q}{p^2}\right\|_{2,p}^2$$

Recall (29), we have

$$\left\|f_\lambda^{\mathrm{I}} - \frac{q}{p}\right\|_{2,p} \le \sqrt{L^2t^2 + 2c\left(D_1\left\|\frac{q}{p^2}\right\|_{W_2^2}^2 t^2 + D_2\frac{\lambda}{c}\left\|\frac{q}{p^2}\right\|_{2,p}^2\right)} \le C_1 t + C_2\lambda^{1/2} \qquad (31)$$

where $C_1 = \sqrt{L^2 + 2cD_1\left\|\frac{q}{p^2}\right\|_{W_s^2}^2}, C_2 = \sqrt{2D_2\left\|\frac{q}{p^2}\right\|_{2,p}^2}$.

Applying Lemma 6, we will have the result for manifold case,

$$\left\|f_\lambda^{\mathrm{I}} - \frac{q}{p}\right\|_{2,p} \le \sqrt{L^2t^{2-2\varepsilon} + 2c\left(D_1\left\|\frac{q}{p^2}\right\|_{W_2^2}^2 t^2 + D_2\frac{\lambda}{c}\left\|\frac{q}{p^2}\right\|_{2,p}^2\right)}$$

$$\le C_1 t^{1-\varepsilon} + C_2\lambda^{1/2} \qquad (32)$$

where $C_1 = \sqrt{L^2 + 2cD_1\left\|\frac{q}{p^2}\right\|_{W_2^2}^2}, C_2 = \sqrt{2D_2\left\|\frac{q}{p^2}\right\|_{2,p}^2}$. $\qquad\square$

### 5.2.2 Bound for Sampling Error

In the next lemma, we will give concentration of the sampling error, $\|f_\lambda^{\mathrm{I}} - f_{\lambda,\mathbf{z}}^{\mathrm{I}}\|_{2,p}$.

**Lemma 9.** *Let $p$ be a density of a probability measure over a domain $X$ and $q$ another density function. They satisfy the assumptions in 4.1. Consider $f_\lambda^{\mathrm{I}}$ and $f_{\lambda,\mathbf{z}}^{\mathrm{I}}$ defined in (8) and (11), with confidence at least $1 - 2e^{-\tau}$, we have*

$$\|f_\lambda^{\mathrm{I}} - f_{\lambda,\mathbf{z}}^{\mathrm{I}}\|_{2,p} \le C_3\left(\frac{\kappa_t\sqrt{\tau}}{\lambda\sqrt{m}} + \frac{k_t\sqrt{\tau}}{\lambda^{7/6}\sqrt{n}}\right)$$

*where $\kappa_t = \sup_{x\in\Omega} k_t(x,x) = \frac{1}{(2\pi t)^{d/2}}$*

*Proof.* Recall that,

$$f_\lambda^{\mathrm{I}} = \arg\min_{f\in H_t}\|\mathcal{K}_{t,p}f - \mathcal{K}_{t,q}\mathbf{1}\|_{2,p}^2 + \lambda\|f\|_{H_t}^2$$

and

$$f_{\lambda,\mathbf{z}}^{\mathrm{I}} = \arg\min_{f\in H_{t,\mathbf{z}}}\frac{1}{n}\sum_{x_i^p\in\mathbf{z}_p}\left((\mathcal{K}_{\mathbf{z}_p}f)(x_i^p) - (\mathcal{K}_{\mathbf{z}_q}\mathbf{1})(x_i^p)\right)^2 + \lambda\|f\|_{H_t}^2$$

Using functional calculus, we will get the explicit formula for $f_\lambda^{\mathrm{I}}$ and $f_{\lambda,\mathbf{z}}^{\mathrm{I}}$ as follows,

$$f_\lambda^{\mathrm{I}} = \left(\mathcal{K}_p^3 + \lambda\mathcal{I}\right)^{-1}\mathcal{K}_p^2\mathcal{K}_q\mathbf{1}$$

and

$$f_{\lambda,\boldsymbol{z}}^{\mathrm{I}} = \left(\mathcal{K}_{\boldsymbol{z}_p}^3 + \lambda\mathcal{I}\right)^{-1} \mathcal{K}_{\boldsymbol{z}_p}^2 \mathcal{K}_{\boldsymbol{z}_q}\mathbf{1}.$$

Then the bound for sampling error is to bound the above two objects. Let $\tilde{f} = \left(\mathcal{K}_{\boldsymbol{z}_p}^3 + \lambda\mathcal{I}\right)^{-1} \mathcal{K}_p^2\mathcal{K}_q\mathbf{1}$. We have $f_\lambda^{\mathrm{I}} - f_{\lambda,\boldsymbol{z}}^{\mathrm{I}} = f_\lambda^{\mathrm{I}} - \tilde{f} + \tilde{f} - f_{\lambda,\boldsymbol{z}}^{\mathrm{I}}$. For $f_\lambda^{\mathrm{I}} - \tilde{f}$, using the fact that $\left(\mathcal{K}_p^3 + \lambda\mathcal{I}\right) f_\lambda^{\mathrm{I}} = \mathcal{K}_p^2\mathcal{K}_q\mathbf{1}$, we have

$$
\begin{aligned}
&f_\lambda^{\mathrm{I}} - \tilde{f} \\
=&f_\lambda^{\mathrm{I}} - \left(\mathcal{K}_{\boldsymbol{z}_p}^3 + \lambda\mathcal{I}\right)^{-1} \left(\mathcal{K}_p^3 + \lambda\mathcal{I}\right) f_\lambda^{\mathrm{I}} \\
=& \left(\mathcal{K}_{\boldsymbol{z}_p}^3 + \lambda\mathcal{I}\right)^{-1} \left(\mathcal{K}_{\boldsymbol{z}_p}^3 - \mathcal{K}_p^3\right) f_\lambda^{\mathrm{I}}
\end{aligned}
$$

And

$$
\begin{aligned}
&\tilde{f} - f_{\lambda,\boldsymbol{z}}^{\mathrm{I}} \\
=& \left(\mathcal{K}_{\boldsymbol{z}_p}^3 + \lambda\mathcal{I}\right)^{-1} \mathcal{K}_p^2\mathcal{K}_q\mathbf{1} - \left(\mathcal{K}_{\boldsymbol{z}_p}^3 + \lambda\mathcal{I}\right)^{-1} \mathcal{K}_{\boldsymbol{z}_p}^2\mathcal{K}_{\boldsymbol{z}_q}\mathbf{1} \\
=& \left(\mathcal{K}_{\boldsymbol{z}_p}^3 + \lambda\mathcal{I}\right)^{-1} \left(\mathcal{K}_p^2\mathcal{K}_q - \mathcal{K}_{\boldsymbol{z}_p}^2\mathcal{K}_{\boldsymbol{z}_q}\right)\mathbf{1}
\end{aligned}
$$

Notice that we have $\mathcal{K}_{\boldsymbol{z}_p}^3 - \mathcal{K}_p^3$ and $\mathcal{K}_{\boldsymbol{z}_p}^2\mathcal{K}_{\boldsymbol{z}_q} - \mathcal{K}_p^2\mathcal{K}_q$ in the identity we get. For these two objects, it is not hard to verify the following identities,

$$
\begin{aligned}
&\mathcal{K}_{\boldsymbol{z}_p}^3 - \mathcal{K}_p^3 \\
=& \left(\mathcal{K}_{\boldsymbol{z}_p} - \mathcal{K}_p\right)^3 + \mathcal{K}_p\left(\mathcal{K}_{\boldsymbol{z}_p} - \mathcal{K}_p\right)^2 + \left(\mathcal{K}_{\boldsymbol{z}_p} - \mathcal{K}_p\right)\mathcal{K}_p\left(\mathcal{K}_{\boldsymbol{z}_p} - \mathcal{K}_p\right) + \left(\mathcal{K}_{\boldsymbol{z}_p} - \mathcal{K}_p\right)^2 \mathcal{K}_p \\
&+ \mathcal{K}_p^2\left(\mathcal{K}_{\boldsymbol{z}_p} - \mathcal{K}_p\right) + \mathcal{K}_p\left(\mathcal{K}_{\boldsymbol{z}_p} - \mathcal{K}_p\right)\mathcal{K}_p + \left(\mathcal{K}_{\boldsymbol{z}_p} - \mathcal{K}_p\right)\mathcal{K}_p^2.
\end{aligned}
$$

And

$$
\begin{aligned}
&\mathcal{K}_{\boldsymbol{z}_p}^2\mathcal{K}_{\boldsymbol{z}_q} - \mathcal{K}_p^2\mathcal{K}_q \\
=& \left(\mathcal{K}_{\boldsymbol{z}_p} - \mathcal{K}_p\right)^2\left(\mathcal{K}_{\boldsymbol{z}_q} - \mathcal{K}_q\right) + \mathcal{K}_p\left(\mathcal{K}_{\boldsymbol{z}_p} - \mathcal{K}_p\right)\left(\mathcal{K}_{\boldsymbol{z}_q} - \mathcal{K}_q\right) + \left(\mathcal{K}_{\boldsymbol{z}_p} - \mathcal{K}_p\right)\mathcal{K}_p\left(\mathcal{K}_{\boldsymbol{z}_q} - \mathcal{K}_q\right) \\
&+ \mathcal{K}_p^2\left(\mathcal{K}_{\boldsymbol{z}_q} - \mathcal{K}_q\right) + \left(\mathcal{K}_{\boldsymbol{z}_p} - \mathcal{K}_p\right)^2\mathcal{K}_q + \mathcal{K}_p\left(\mathcal{K}_{\boldsymbol{z}_p} - \mathcal{K}_p\right)\mathcal{K}_q + \left(\mathcal{K}_{\boldsymbol{z}_p} - \mathcal{K}_p\right)\mathcal{K}_p\mathcal{K}_q.
\end{aligned}
$$

Thus, in these two identities, the only two random variables are $\mathcal{K}_{\boldsymbol{z}_p} - \mathcal{K}_p$ and $\mathcal{K}_{\boldsymbol{z}_q} - \mathcal{K}_q$. By results about concentration of $\mathcal{K}_{\boldsymbol{z}_p}$ and $\mathcal{K}_{\boldsymbol{z}_q}$, we have with probability $1 - 2e^{-\tau}$,

$$
\begin{aligned}
&\left\|\mathcal{K}_{\boldsymbol{z}_p} - \mathcal{K}_p\right\|_{\mathcal{H}\to\mathcal{H}} \leq \frac{\kappa_t\sqrt{\tau}}{\sqrt{n}}, \\
&\left\|\mathcal{K}_{\boldsymbol{z}_q} - \mathcal{K}_q\right\|_{\mathcal{H}\to\mathcal{H}} \leq \frac{\kappa_t\sqrt{\tau}}{\sqrt{m}}, \\
&\left\|\mathcal{K}_{\boldsymbol{z}_q}\mathbf{1} - \mathcal{K}_q\mathbf{1}\right\|_{\mathcal{H}} \leq \frac{\kappa_t\sqrt{2\tau}}{\sqrt{m}}
\end{aligned}
\tag{33}
$$

And we know that for a large enough constant $c$ which is independent of $t$ and $\lambda$,

$$\|\mathcal{K}_p\|_{\mathcal{H}\to\mathcal{H}} < c, \|\mathcal{K}_q\|_{\mathcal{H}\to\mathcal{H}} < c, \left\|\left(\mathcal{K}_{\boldsymbol{z}_p}^3 + \lambda\mathcal{I}\right)^{-1}\right\|_{\mathcal{H}\to\mathcal{H}} \leq \frac{1}{\lambda}, \|\mathcal{K}_q\mathbf{1}\|_{\mathcal{H}} < c$$

and

$$\|f_\lambda^{\mathrm{I}}\|_{\mathcal{H}}^2 = \sum_i \frac{\sigma_i^5}{(\sigma_i^3 + \lambda)^2}\left\langle \frac{q}{p}, u_i\right\rangle^2 \leq \left(\sup_{\sigma>0} \frac{\sigma^5}{(\sigma^3 + \lambda)^2}\right)\sum_i \left\langle \frac{q}{p}, u_i\right\rangle^2 \leq c^2 \frac{1}{\lambda^{1/3}}\left\|\frac{q}{p}\right\|_{2,p}^2$$

thus, $\|f_\lambda^{\mathrm{I}}\|_{\mathcal{H}} \leq \frac{c}{\lambda^{1/6}}\left\|\frac{q}{p}\right\|_{2,p}$.

Notice that $\left\|(\mathcal{K}_{z_p} - \mathcal{K}_p)^2\right\|_{\mathcal{H}} \leq \|\mathcal{K}_{z_p} - \mathcal{K}_p\|_{\mathcal{H}}^2$ and $\left\|(\mathcal{K}_{z_p} - \mathcal{K}_p)^3\right\|_{\mathcal{H}} \leq \|\mathcal{K}_{z_p} - \mathcal{K}_p\|_{\mathcal{H}}^3$, both of this could be of smaller order compared with $\|\mathcal{K}_{z_p} - \mathcal{K}_p\|_{\mathcal{H}}$. For simplicity we hide the term including them in the final bound without changing the dominant order. We could also hide the terms with the product of any two the random variables in Eq. 38, which is of prior order compared to the term with only one random variable. Now let us put everything together,

$$\|f_\lambda^{\mathrm{I}} - f_{\lambda,z}^{\mathrm{I}}\|_{2,p} \leq c^{1/2}\|f_\lambda^{\mathrm{I}} - f_{\lambda,z}^{\mathrm{I}}\|_{H_t}$$

$$\leq c^{1/2}\left(\frac{c^3 \kappa_t \sqrt{\tau}}{\lambda^{7/6}\sqrt{n}}\left\|\frac{q}{p}\right\|_{2,p} + \frac{c^2 \kappa_t \sqrt{\tau}}{\lambda\sqrt{m}}\right)$$

$$\leq C_3\left(\frac{\kappa_t \sqrt{\tau}}{\lambda\sqrt{m}} + \frac{\kappa_t \sqrt{\tau}}{\lambda^{7/6}\sqrt{n}}\right)$$

where $C_3 = c^{5/2}\max\left(c\left\|\frac{q}{p}\right\|_{2,p}, 1\right)$. $\qquad\square$

# 6  Experiments

In this section we explore the empirical performance of our methods under various settings. We will primarily concentrate on our setting Type II and use the same Gaussian kernel for the integral operator and the regularization term to simplify model selection.

We start by proposing an unsupervised technique for parameter selection (CD-CV for Cross-Density Cross-Validation) based on a natural performance measure unique to our setting and briefly describe the data sets and the re-sampling procedures we use. We proceed to evaluate our method on several datasets, including simulated examples and real-world data sets.

## 6.1  Experimental Setting and Model Selection

**The setting.** In our experiments, we have a set of a data set $X^p = \{x_i^p, i = 1, 2, \ldots, n\}$ and another set of instances $X^q = \{x_j^q, j = 1, 2, \ldots, m\}$. The goal is to estimate $\frac{q}{p}$ under the assumption that $X^p$ is sampled from $p$ and $X^q$ is sampled from $q$.

We note that our algorithms typically has two parameters, which need to be selected, the kernel width $t$ and the regularization parameter $\lambda$. In general choosing parameters in a unsupervised or semi-supervised setting is a hard problem as it may be difficult to validate the resulting classifier/estimator. However, certain features of our setting allow us to construct an adequate unsupervised proxy for the performance of the algorithm. Now we construct a performance measure for the quality of the estimator.

**Performance Measures and CD-CV Model Selection.** We describe a set of performance measures to use for parameter selection.

For a given function $u$, we have the following importance sampling equality (Eq. 1):

$$\mathbb{E}_q(u(x)) = \mathbb{E}_p\left(u(x)\frac{q(x)}{p(x)}\right).$$

If $f(x)$ is an approximation of the true ratio $\frac{q}{p}$, using the samples from $X^p$ and $X^q$ respectively, we will have the following approximation to the above equation:

$$\frac{1}{n}\sum_{i=1}^{n} u(x_i^p)f(x_i^p) \approx \frac{1}{m}\sum_{j=1}^{m} u(x_j^q).$$

Therefore, after obtaining an estimate $f$ of the ratio, we can validate it by using a set of test functions $U = \{u_1, u_2, \ldots, u_F\}$ using the following performance measure:

$$J(f; X^p, X^q, U) = \frac{1}{F}\sum_{l=1}^{F}\left(\sum_{i=1}^{n} u_l(x_i^p)f(x_i^p) - \sum_{j=1}^{m} u_l(x_j^q)\right)^2 \qquad (34)$$

where $U = \{u_1, u_2, \ldots, u_F\}$ is a collection of function chosen as criterion. Using this performance measure allows various cross-validation procedures to be sued for parameter selection.

We note that this way of measuring the error is related to the LSIF [14] and KLIEP [31], algorithms. However, there a similar measure is used to construct an approximation to the ratio $frac{q}{p}$ using functions $u_1, \ldots, u_F$ as a basis. In our setting, to choose parameters, we can use validations sets (such as linear functions) which are poorly suited as a basis for approximating the density ratio.

**Choice of validation function sets for parameter selection.** In principle, any set of (sufficiently well-behaved) functions can be used as a validation set. From a practical point of view we would like functions to be simple to compute and readily available for different data sets.

In the our experiments, we will use the following two families of functions for parameter tuning:

(1) Sets of random linear functions $u(x) = \beta^T x$ where $\beta \sim N(0, Id)$.
(2) Sets of random half-space indicator functions, $u(x) = \mathbf{1}_{\beta^T x > 0}$ where $\beta \sim N(0, Id)$.

**Remark 1:** We have also tried (a) coordinates functions, (b) random combination of kernel functions, and (c) random combination of kernel functions with thresholding. In our experience the coordinate functions are not rich enough for adequate parameter tuning. On the other hand, using the kernel functions significantly increases the complexity of the procedure (due to the necessity of choosing the kernel width and other parameters) without increasing the performance significantly.

**Remark 2:** Note that for linear functions, the cardinality of the set should not exceed the dimension of the space due to linear dependence.

**Remark 3:** It appears that linear functions work well for regression tasks, while half-spaces are well-suited for classification.

**Procedures for parameter selection.**

We optimize the performance using cross-validation by splitting the data set in two parts $X^{p,train}$ and $X^{q,train}$ used for training and $X^{p,cv}$ and $X^{q,cv}$ used for validation, and repeating this process five times to find the optimal values of parameters[8].

For the two parameters which need to be tuned, the kernel width $t$ and the regularization parameter $\lambda$, we specify a parameter grid as follows. The range for kernel width $t$ is $(t_0, 2t_0, \ldots, 2^9 t_0)$, where $t_0$ is the average distance of the 10 nearest neighbors, and regularization parameter $\lambda$ is $(1e-5, 1e-6, \ldots, 1e-10)$.

## 6.2 Data sets and Resampling

In our experiments, several data sets are considered: Bank8FM, CPUsmall and Kin8nm for regression; and USPS and 20 news groups for classification.

For each data set, we assume they are i.i.d. sampled from a distribution denoted by $p$. We draw the first 500 or 1000 points from the original data set as $X^p$. To obtain $X^q$, we apply a resampling scheme on the remaining points of the original data set. Two ways of resampling, using the features of the data and using the label information, are used (along the lines similar to those proposed in [9]).

Specifically, given a set of data points with labels $\{(x_1, y_1), (x_2, y_2), \ldots, (x_n, y_n)\}$ we resample as follows:

- **Resampling using feature information (labels $y_i$ are not used).** We subsample the data points so that the probability $P_i$ of selecting the instance $i$, is defined by the following (sigmoid) function:

$$P_i = \frac{e^{(a\langle x_i, e_1\rangle - b)/\sigma_v}}{1 + e^{(a\langle x_i, e_1\rangle - b)/\sigma_v}}$$

  where $a, b$ are the resampling parameters, $e_1$ is the first principal component, and $\sigma_v$ is the standard deviation of the projection to $e_1$. Note that in this resampling scheme, the

probability of taking one point is only conditioned on the feature information $x_i$. This resampling method will be denoted by PCA$(a, b)$.

- **Resampling using label information.** The probability of selecting the $i$'th instance, denoted by $P_i$, is defined by

$$P_i = \begin{cases} 1 & y_1 \in L_q \\ 0 & \text{Otherwise.} \end{cases}$$

where $y_i \in L = \{1, 2, \ldots, k\}$ and $L_q$ is a subset of the complete label set $L$. We apply this for binary problems obtained by aggregating different classes in the multi-class setting.

## 6.3 Testing the FIRE algorithm

In first experiment, we test our method for selecting the parameters, which is described in Section 6.1, by focusing on the the error $J(f; X^p, X^q, U)$ in Eq. 34 for different function classes $U$. We use different families of functions for tuning parameters and validation. This measure is important because in practice the functions we are interested may not be in the collection we chosen for validation. To avoid confusion, we denote the function for cross validation by $f^{\text{cv}}$ and the function for measuring error by $f^{\text{err}}$.

We use the CPUsmall and USPS hand-written digits data sets. For each of them, we generate two data sets $X^p$ and $X^q$ using the resampling method, PCA$(a, \sigma_v)$, describe in Section 6.2. We compare FIRE with several methods including TIKDE, LSIF. Figure 1 gives an illustration of the procedure and usage of data for the experiments. And the results are shown in Table 1 and 2. The numbers in the table are the average errors defined in Eq. 34 on held-out set $X^{\text{err}}$ over 5 trials, using different criterion functions $f^{\text{cv}}$(Columns) and error-measuring functions $f^{\text{err}}$(Row). $N$ is the number of random function we are using for the cross-validation.

Figure 1: First of all $X^p, X^q$ are split into $X^{p,\text{cv}}$ and $X^{p,\text{err}}$, $X^{q,\text{cv}}$ and $X^{q,\text{err}}$. Then we further split $X^{p,\text{cv}}$ into $k$ folds. For each fold $i$, density ratios at the sample points are estimated using only folds $j \neq i$ and $X^{q,\text{cv}}$, and compute the error using fold $i$ and $X^{q,\text{cv}}$. We choose the parameter gives the best average error over the $k$ folds of $X^{p,\text{cv}}$. And we measure the final performance using $X^{p,\text{err}}$ and $X^{q,\text{err}}$.

For the error-measuring functions, we have several choices as follows:

(1) Linear: Sets of Random linear functions $f(x) = \beta^T x$ where $\beta \sim N(0, Id)$.

(2) Half-space: Sets of random half-space indicator functions, $f(x) = \mathbf{1}_{\beta^T x > 0}$ where $\beta \sim N(0, Id)$.

(3) Kernel: Sets of random linear combination of kernel functions centered at the training data, $f(x) = \gamma^T K$ where $\gamma \sim N(0, Id)$ and $K_{ij} = k(x_i, x_j)$ where $x_i$ are points from the data set.

(4) K-indicator: Sets of random kernel indicator functions centered at the training data, $f = \mathbf{1}_{\gamma^T K > 0}$ where $\gamma \sim N(0, Id)$ and $K_{ij} = k(x_i, x_j)$ where $x_i$ are points from the data set.

(5) Coord: Sets of coordinate functions.

Table 1: USPS data set with resampling using PCA$(5, \sigma_v)$, where $\sigma_v$ is the standard deviation of projected value on the first principal component. And $|X^p| = 500$ and $|X^q| = 1371$. Around 400 in $X^p$ and 700 in $X^q$ are used in 5-folds CV.

| | | Linear | | Half-spaces | |
|---|---|---|---|---|---|
| | | N=50 | N=200 | N=50 | N=200 |
| Linear | TIKDE | 10.9 | 10.9 | 10.9 | 10.9 |
| | LSIF | 14.1 | 14.1 | 26.8 | 28.2 |
| | FIRE($L_{2,p}$) | **3.56** | 3.75 | 5.52 | 6.32 |
| | FIRE($L_{2,p} + L_{2,q}$) | 4.66 | 4.69 | 7.35 | 6.82 |
| | FIRE($L_{2,q}$) | 5.89 | 6.24 | 9.28 | 9.28 |
| Half-spaces | TIKDE | 0.0259 | 0.0259 | 0.0259 | 0.0259 |
| | LSIF | 0.0388 | 0.0388 | 0.037 | 0.039 |
| | FIRE($L_{2,p}$) | 0.00966 | **0.0091** | 0.0103 | 0.0118 |
| | FIRE($L_{2,p} + L_{2,q}$) | 0.0094 | 0.0102 | 0.0143 | 0.0107 |
| | FIRE($L_{2,q}$) | 0.0124 | 0.0135 | 0.0159 | 0.0159 |
| Kernel | TIKDE | 4.74 | 4.74 | 4.74 | 4.74 |
| | LSIF | 16.1 | 16.1 | 15.6 | 13.8 |
| | FIRE($L_{2,p}$) | 1.19 | **1.05** | 2.78 | 3.57 |
| | FIRE($L_{2,p} + L_{2,q}$) | 2.06 | 1.99 | 4.2 | 2.59 |
| | FIRE($L_{2,q}$) | 5.16 | 4.27 | 6.11 | 6.11 |
| K-Indicator | TIKDE | 0.0415 | 0.0415 | 0.0415 | 0.0415 |
| | LSIF | 0.0435 | 0.0435 | 0.0531 | 0.044 |
| | FIRE($L_{2,p}$) | 0.00862 | 0.00676 | 0.0115 | 0.0114 |
| | FIRE($L_{2,p} + L_{2,q}$) | **0.00559** | 0.00575 | 0.0191 | 0.0108 |
| | FIRE($L_{2,q}$) | 0.0117 | 0.00935 | 0.0217 | 0.0217 |
| Coord. | TIKDE | 0.0541 | 0.0541 | 0.0541 | 0.0541 |
| | LSIF | 0.0647 | 0.0647 | 0.139 | 0.162 |
| | FIRE($L_{2,p}$) | 0.0183 | **0.0165** | 0.032 | 0.0334 |
| | FIRE($L_{2,p} + L_{2,q}$) | 0.0211 | 0.0201 | 0.0423 | 0.0355 |
| | FIRE($L_{2,q}$) | 0.0277 | 0.0233 | 0.0496 | 0.0496 |

Table 2: CPUsmall data set with resampling using PCA$(5, \sigma_v)$, where $\sigma_v$ is the standard deviation of projected value on the first principal component. And $|X^p| = 1000$ and $|X^q| = 2000$. Around 800 in $X^p$ and 1000 in $X^q$ are used in 5-folds CV.

| | | Linear | | Half-spaces | |
|---|---|---|---|---|---|
| | | N=50 | N=200 | N=50 | N=200 |
| Linear | TIKDE | 0.102 | 0.0965 | 0.102 | 0.0984 |
| | LSIF | 0.115 | 0.115 | 0.115 | 0.115 |
| | FIRE($L_{2,p}$) | 0.0908 | 0.0858 | 0.0891 | 0.0924 |
| | FIRE($L_{2,p} + L_{2,q}$) | 0.0832 | 0.0825 | 0.0825 | **0.0718** |
| | FIRE($L_{2,q}$) | 0.0889 | 0.0907 | 0.0932 | 0.0899 |
| Half-spaces | TIKDE | 0.00469 | 0.00416 | 0.00469 | 0.00462 |
| | LSIF | 0.00487 | 0.00487 | 0.00487 | 0.00487 |
| | FIRE($L_{2,p}$) | 0.00393 | 0.00389 | 0.00435 | 0.00436 |
| | FIRE($L_{2,p} + L_{2,q}$) | 0.00385 | 0.00383 | 0.00383 | **0.00345** |
| | FIRE($L_{2,q}$) | 0.00421 | 0.0044 | 0.00459 | 0.00427 |
| Kernel | TIKDE | 9.82 | 8.48 | 9.82 | 9.3 |
| | LSIF | 9.6 | 9.6 | 9.6 | 9.6 |
| | FIRE($L_{2,p}$) | 6.96 | 6.17 | 8.02 | 8.19 |
| | FIRE($L_{2,p} + L_{2,q}$) | 6.62 | 6.62 | 6.62 | **6.35** |
| | FIRE($L_{2,q}$) | 7.23 | 7.17 | 7.44 | 7.38 |
| K-Indicator | TIKDE | 0.00411 | 0.00363 | 0.00411 | 0.00404 |
| | LSIF | 0.00478 | 0.00478 | 0.00478 | 0.00478 |
| | FIRE($L_{2,p}$) | 0.0033 | 0.00313 | 0.0036 | 0.00373 |
| | FIRE($L_{2,p} + L_{2,q}$) | 0.00306 | 0.00306 | 0.00306 | **0.00288** |
| | FIRE($L_{2,q}$) | 0.00358 | 0.00354 | 0.00365 | 0.00366 |
| Coord. | TIKDE | 0.00784 | 0.0077 | 0.00784 | 0.00758 |
| | LSIF | 0.00774 | 0.00774 | 0.00774 | 0.00774 |
| | FIRE($L_{2,p}$) | 0.00696 | 0.00676 | 0.00681 | 0.00734 |
| | FIRE($L_{2,p} + L_{2,q}$) | 0.00647 | 0.00637 | 0.00637 | **0.00584** |
| | FIRE($L_{2,q}$) | 0.00693 | 0.00692 | 0.00699 | 0.00689 |

## 6.4 Supervised Learning: Regression and Classification

In our experiments, we compare our method FIRE with several methods under the setting of supervised learning, i.e. regression and classification. More specifically, we consider the situation part or all of the training set $X^p$ are labeled and all of $X^q$ are unlabeled. In the following experiments, we will estimate the density ratio function using 1000 points in $X^p$ and use the labeled data from $X^p$ to build a regression function or classifier on $X^q$.

### 6.4.1 Regression

Given data sets $(X^p, Y^p)$ where $X^p$ is for independent variable, and $Y^p$ is for dependent variable, and a test data set $X^q$ with a different distribution, the regression problem is to obtain a function a predictor on $X^q$. To make the comparison between unweighted regression method and different weighting schemes, we use the simplest regression method, the least square linear regression. With this method, the regression function is of the form

$$f(x, \beta) = \beta^t x,$$

where $\beta = (XWX^T)^+ XWY$ and $(\cdot)^+$ denotes the pseudo-inverse of a matrix. Here $W$ is a diagonal matrix with the estimated density ratio on the diagonal. These are estimated using FIRE and other density ratio estimation methods for comparison. The results on 3 regression data sets are shown in Table 5, 3 and 4.

Table 3: CPUsmall resampled using $\mathrm{PCA}(5, \sigma_v)$, where $\sigma_v$ is the standard deviation of projected value on the first principal component. $|X^p| = 1000$, $|X^q| = 2000$.

| No. of Labeled | 100 | | 200 | | 500 | | 1000 | |
|---|---|---|---|---|---|---|---|---|
| Weighting method | Linear | Half-spaces | Linear | Half-spaces | Linear | Half-spaces | Linear | Half-spaces |
| OLS | 0.740 | | 0.497 | | 0.828 | | 0.922 | |
| TIKDE | 0.379 | 0.359 | 0.299 | 0.291 | 0.278 | 0.279 | 0.263 | 0.267 |
| KMM | 1.857 | 1.857 | 1.899 | 1.899 | 2.508 | 2.508 | 2.739 | 2.739 |
| LSIF | 0.390 | 0.390 | 0.309 | 0.309 | 0.329 | 0.329 | 0.314 | 0.314 |
| FIRE($L_{2,p}$) | 0.327 | 0.327 | 0.286 | 0.286 | 0.272 | 0.272 | 0.260 | 0.260 |
| FIRE($L_{2,p} + L_{2,q}$) | 0.326 | 0.330 | 0.285 | 0.287 | 0.272 | 0.272 | 0.261 | **0.259** |
| FIRE($L_{2,q}$) | **0.324** | 0.333 | **0.284** | 0.288 | **0.271** | 0.272 | 0.261 | 0.260 |

Table 4: Kin8nm resampled using $\mathrm{PCA}(10, \sigma_v)$. $|X^p| = 1000$, $|X^q| = 2000$.

| No. of Labeled | 100 | | 200 | | 500 | | 1000 | |
|---|---|---|---|---|---|---|---|---|
| Weighting method | Linear | Half-spaces | Linear | Half-spaces | Linear | Half-spaces | Linear | Half-spaces |
| OLS | 0.588 | | 0.552 | | 0.539 | | 0.535 | |
| TIKDE | 0.572 | 0.574 | 0.545 | 0.545 | 0.526 | 0.529 | 0.523 | 0.524 |
| KMM | 0.582 | 0.582 | 0.547 | 0.547 | 0.522 | 0.522 | **0.514** | **0.514** |
| LSIF | 0.565 | 0.563 | 0.543 | 0.541 | 0.520 | 0.520 | 0.517 | 0.516 |
| FIRE($L_{2,p}$) | 0.567 | **0.560** | 0.548 | **0.540** | 0.524 | **0.519** | 0.522 | 0.515 |
| FIRE($L_{2,p} + L_{2,q}$) | 0.563 | **0.560** | 0.546 | **0.540** | 0.522 | **0.519** | 0.520 | 0.515 |
| FIRE($L_{2,q}$) | 0.563 | **0.560** | 0.546 | 0.541 | 0.522 | **0.519** | 0.520 | 0.515 |

Table 5: Bank8FM resampled using $\mathrm{PCA}(1, \sigma_v)$. $|X^p| = 1000$, $|X^q| = 2000$.

| No. of Labeled | 100 | | 200 | | 500 | | 1000 | |
|---|---|---|---|---|---|---|---|---|
| Weighting method | Linear | Half-spaces | Linear | Half-spaces | Linear | Half-spaces | Linear | Half-spaces |
| OLS | 0.116 | | 0.111 | | 0.105 | | 0.101 | |
| TIKDE | 0.111 | 0.111 | **0.100** | **0.100** | **0.096** | **0.096** | **0.092** | **0.092** |
| KMM | 0.112 | 0.161 | 0.103 | 0.164 | 0.099 | 0.180 | 0.095 | 0.178 |
| LSIF | 0.113 | 0.113 | 0.109 | 0.109 | 0.104 | 0.104 | 0.099 | 0.099 |
| FIRE($L_{2,p}$) | **0.110** | **0.110** | 0.101 | 0.102 | 0.097 | 0.097 | 0.093 | 0.094 |
| FIRE($L_{2,p} + L_{2,q}$) | 0.113 | **0.110** | 0.103 | 0.102 | 0.099 | 0.097 | 0.097 | 0.094 |
| FIRE($L_{2,q}$) | 0.112 | 0.118 | 0.102 | 0.106 | 0.099 | 0.103 | 0.096 | 0.102 |

### 6.4.2 Classification

Similarly to the case of regression the density ratio can also be used for building a classifier such as SVM. Given a set of labeled data, $\{(x_1, y_1), (x_2, y_2), \ldots, (x_n, y_n)\}$ and $x_i \sim q$, we building a linear classifier $f$ by the weighted linear SVM algorithm as follows:

$$f = \arg\min_{\beta \in R^d} \frac{C}{n} \sum_{i=1}^{n} w_i (\beta^T x_i - y_i)_+ + \|\beta\|_2^2$$

The weights $w_i$'s are obtained by various density ratios estimation algorithms using two data sets $X^p$ and $X^q$. Note that estimating the density ratios using $X^p$ and $X^q$ is completely independent of the label information. We also explore the performance of these weighted SVM as the number of labeled points used for training classifier changes. In the experiments, we first estimate the density ratios on the whole $X^p$ with the parameters selected by cross validation. Then we subsample a portion of $X^p$ and use their labels to train the classifier. And the performance of the classifier in terms of prediction error is estimated using all the points in $X^q$. The results on USPS hand-written digits and 20 news groups are shown in Table 6, 7, 8 and 9.

Table 6: USPS resampled using **Feature** information, PCA$(5, \sigma_v)$, where $\sigma_v$ is the standard deviation of projected value on the first principal component. $|X^p| = 1000$ and $|X^q| = 1371$, with $0 - 4$ as $-1$ class and $5 - 9$ as $+1$ class.

| No. of Labeled | 100 | | 200 | | 500 | | 1000 | |
|---|---|---|---|---|---|---|---|---|
| Weighting method | Linear | Half-spaces | Linear | Half-spaces | Linear | Half-spaces | Linear | Half-spaces |
| SVM | 0.102 | | 0.081 | | 0.057 | | 0.058 | |
| TIKDE | 0.094 | 0.094 | 0.072 | 0.072 | 0.049 | 0.049 | 0.042 | 0.042 |
| KMM | 0.081 | 0.081 | 0.059 | 0.059 | 0.047 | 0.047 | 0.044 | 0.044 |
| LSIF | 0.095 | 0.102 | 0.073 | 0.081 | 0.050 | 0.057 | 0.044 | 0.058 |
| FIRE($L_{2,p}$) | 0.089 | 0.068 | 0.053 | 0.050 | **0.041** | **0.041** | 0.037 | 0.036 |
| FIRE($L_{2,p} + L_{2,q}$) | 0.070 | 0.070 | 0.051 | 0.051 | **0.041** | **0.041** | 0.036 | 0.036 |
| FIRE($L_{2,q}$) | **0.055** | 0.073 | **0.048** | 0.054 | **0.041** | 0.044 | **0.034** | 0.039 |

Table 7: USPS resampled based on **Label** information, $X^q$ only contains point with labels in $L' = \{0, 1, 5, 6\}$. The binary classes are with $+1$ class$= \{0, 1, 2, 3, 4\}$, $-1$ class$= \{5, 6, 7, 8, 9\}$. And $|X^p| = 1000$ and $|X^q| = 2000$.

| No. of Labeled | 100 | | 200 | | 500 | | 1000 | |
|---|---|---|---|---|---|---|---|---|
| Weighting method | Linear | Half-spaces | Linear | Half-spaces | Linear | Half-spaces | Linear | Half-spaces |
| SVM | 0.186 | | 0.164 | | 0.129 | | 0.120 | |
| TIKDE | 0.185 | 0.185 | 0.164 | 0.164 | 0.124 | 0.124 | 0.105 | 0.105 |
| KMM | **0.175** | **0.175** | **0.135** | **0.135** | **0.103** | **0.103** | **0.085** | **0.085** |
| LSIF | 0.185 | 0.185 | 0.162 | 0.163 | 0.122 | 0.122 | 0.108 | 0.108 |
| FIRE($L_{2,p}$) | 0.179 | 0.184 | 0.161 | 0.161 | 0.115 | 0.120 | 0.107 | 0.105 |
| FIRE($L_{2,p} + L_{2,q}$) | 0.180 | 0.185 | 0.161 | 0.162 | 0.116 | 0.120 | 0.106 | 0.107 |
| FIRE($L_{2,q}$) | 0.183 | 0.184 | 0.160 | 0.162 | 0.118 | 0.120 | 0.106 | 0.103 |

Table 8: 20 News groups resampled using **Feature** information, PCA$(5, \sigma_v)$, where $\sigma_v$ is the standard deviation of projected value on the first principal component. $|X^p| = 1000$ and $|X^q| = 1536$, with $\{2, 4, \ldots, 20\}$ as $-1$ class and $\{1, 3, \ldots, 19\}$ as $+1$ class.

| No. of Labeled | 100 | | 200 | | 500 | | 1000 | |
|---|---|---|---|---|---|---|---|---|
| Weighting method | Linear | Half-spaces | Linear | Half-spaces | Linear | Half-spaces | Linear | Half-spaces |
| SVM | 0.326 | | 0.286 | | 0.235 | | 0.204 | |
| TIKDE | 0.326 | 0.326 | 0.286 | 0.285 | 0.235 | 0.235 | 0.204 | 0.204 |
| KMM | 0.338 | 0.338 | 0.303 | 0.303 | 0.252 | 0.252 | 0.242 | 0.242 |
| LSIF | 0.329 | 0.325 | 0.297 | 0.285 | 0.238 | 0.235 | 0.210 | 0.204 |
| FIRE($L_{2,p}$) | **0.314** | 0.324 | 0.276 | 0.278 | **0.231** | 0.234 | 0.202 | 0.210 |
| FIRE($L_{2,p} + L_{2,q}$) | 0.315 | 0.323 | 0.276 | 0.277 | 0.232 | 0.233 | 0.200 | 0.208 |
| FIRE($L_{2,q}$) | 0.317 | 0.321 | 0.277 | **0.275** | 0.232 | **0.231** | **0.197** | 0.207 |

Table 9: 20 News groups resampled based on **Label** information, $X^q$ only contains point with labels in $L' = \{1, 2, \ldots, 8\}$. The binary classes are with $+1$ class$= \{1, 2, 3, 4\}$, $-1$ class$= \{5, 6, \ldots, 20\}$. $|X^p| = 1000$ and $|X^q| = 4148$.

| No. of Labeled | 100 | | 200 | | 500 | | 1000 | |
|---|---|---|---|---|---|---|---|---|
| Weighting method | Linear | Half-spaces | Linear | Half-spaces | Linear | Half-spaces | Linear | Half-spaces |
| SVM | 0.354 | | 0.333 | | 0.300 | | 0.284 | |
| TIKDE | 0.354 | 0.353 | 0.334 | 0.335 | 0.299 | 0.298 | 0.281 | 0.285 |
| KMM | 0.368 | 0.368 | 0.341 | 0.341 | **0.295** | **0.295** | **0.270** | **0.270** |
| LSIF | 0.353 | 0.354 | 0.336 | 0.334 | 0.304 | 0.305 | 0.286 | 0.284 |
| FIRE($L_{2,p}$) | **0.347** | 0.348 | 0.334 | 0.332 | 0.303 | 0.300 | 0.282 | 0.277 |
| FIRE($L_{2,p} + L_{2,q}$) | 0.348 | 0.348 | 0.332 | 0.332 | 0.301 | 0.301 | 0.277 | 0.277 |
| FIRE($L_{2,q}$) | **0.347** | 0.349 | **0.330** | **0.330** | 0.303 | 0.300 | 0.284 | 0.278 |

## 6.5 Simulated Examples

### 6.5.1 Simulated Dataset 1.

We use a simple example, where the two densities are known, to demonstrate the properties of our methods and how the number of data points influences the performance.

For this experiment, we suppose $p = 0.5N(-2, 1^2) + 0.5N(2, 0.5^2)$ and $q = N(0, 0.5^2)$ and fix $|X^q| = 2000$, and vary $|X_p|$ from 50 to 1000. We compare our method with the other two methods for the same problem: TIKDE and KMM. For all the methods we consider in this experiment, we will choose the optimal parameter based on the empirical $L_2$ norm of the difference between the estimated ratio and the true ratio, which is supposed to be known in this toy example. Figure 2 gives the reader an intuition about how the estimated ratios behave for different methods.

| (a) TIKDE | (b) FIRE | (c) KMM |
|---|---|---|

Figure 2: Plots of estimation of the ratio of densities with $|X^p| = 500$ of points from $p = 0.5N(-2, 1^2) + 0.5N(2, 0.5^2)$ and $|X^q| = 2000$ points from $q = N(0, 0.5^2)$. The blues lines are true ratio, $\frac{q}{p}$. Left column is the estimations from KDE with proper chosen threshold. Middle column is the estimations from our method, FIRE. And right one is the estimation from KMM.

And Figure 3 shows how different methods perform when $|X^p|$ varies from 50 to 1000 and $|X^q|$ is fixed to be 2000. The boxplot is also a good way to illustrate the stability of the methods over 50 independent repetitions.

Figure 3: Number of points from $p$, $n$ varies from 50 to 1000 as the horizontal axis indicates, and the number of points from $q$ is fixed to be 2000. For each $n$, the three bars, from left to right, belongs to TIKDE, FIRE(marked as red) and KMM.

### 6.5.2 Simulated Dataset 2.

In the second simulated example, we will test our method for various kernels and different norms as the cost function. More specifically, we suppose $p = N(0, 0.5^2)$ and $q = \text{Unif}([-1, 1])$. We will use this example to explore the power of our methods with different kernels. Three settings are considered in this experiments: (1)Different kernels $k_h$ for the RKHS. We use polynomial kernels of degree 1, 5 and 20, exponential kernel and Gaussian kernel; (2) Type-I setting and Type-II setting; (3) Different norm for the cost function in the algorithm, i.e. $\| \cdot \|_{2,p}$ and $\| \cdot \|_{2,q}$. In this example, $\| \cdot \|_{2,p}$ focuses on the region close 0, but still has penalty outside interval $[-1, 1]$; $\| \cdot \|_{2,p}$ has uniform penalty on $[-1, 1]$ and has no penalty at all outside the interval.

In all settings, we fix the convolution kernel to be Gaussian, $k_t$. When the RKHS kernel is exponential and Gaussian, we also need to decide their width. For simplicity, we just fix their width to be $20t$, where $t$ is the width of the convolution kernel $k_t$. For setting Type-I, we will set $|X^p| = 500$ and $|X^q| = 500$; for Type-II setting, we only specify $|X^p| = 500$. The results are shown in Figure 4.

## Footnotes

[1]In fact our formulation is quite close to the original formulation of Tikhonov.

[2]This could be important in various sampling procedures, for example, when the normalizing coefficients are hard to estimate.

[3]Obtained by dividing the standard kernel density estimator for $q$ by a thresholded kernel density estimator for $p$ Interestingly, despite its simplicity it performs quite well in many settings.

[4]In particular, this becomes an issue for model selection, see Section 6.

[5] $f$ needs to be in a function class where point evaluations are defined.

[6] Especially in Eq. 7, where the identity has an error term depending on $t$.

[7]Strictly speaking, an arbitrary matrix can only be reduced to the Jordan canonical form, but an arbitrarily small perturbation of any matrix can be diagonalized over the complex numbers.

[8]We note that this procedure cannot be used with KMM as it has no out-of-sample extension. Therefore in subsection 6.3 we do not compare our method with KMM since there is no obvious way to extend the results to the validation data set.

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

# A  Proof for Lemma 5

*Proof.* RKHS is unique for a given domain and kernel, so is independent of the measure used to define the $L_{2,\rho}$. Thus for any $g \in L_{2,p}$, there should be $h \in L_2$ such that $\mathcal{K}_{t,p}^{1/2} g = \mathcal{K}_t^{1/2} h$ and

$$\|g\|_{2,p} = \|\mathcal{K}_{t,p}^{1/2} g\|_{H_t} = \|\mathcal{K}_t^{1/2} h\|_{H_t} = \|h\|_2.$$

Since this is true for arbitrary $g \in L_{2,p}$, we have

$$\min_{g \in L_{2,p}} \left\| f - \mathcal{K}_{t,p}^{1/2} g \right\|_{2,p}^2 + \lambda' \|g\|_{2,p}^2 = \min_{h \in L_2} \left\| f - \mathcal{K}_t^{1/2} h \right\|_{2,p}^2 + \lambda' \|h\|_2^2$$

Because $\| \cdot \|_{2,p} \leq \Gamma \| \cdot \|_2$,

$$\min_{h \in L_2} \left\| f - \mathcal{K}_t^{1/2} h \right\|_{2,p}^2 + \lambda' \|h\|_2^2 \leq \Gamma^2 \left( \min_{h \in L_2} \left\| f - \mathcal{K}_t^{1/2} h \right\|_2^2 + \frac{\lambda'}{\Gamma^2} \|h\|_2^2 \right) \qquad (35)$$

To bound

$$\min_{h \in L_2} \left\| f - \mathcal{K}_t^{1/2} h \right\|_2^2 + \frac{\lambda'}{\Gamma^2} \|h\|_2^2$$

We need the Fourier transform $F : L_2(\mathbb{R}^d) \rightarrow L_2(\mathbb{R}^d)$, defined as

$$\hat{f}(\xi) = \int_{\mathbb{R}^d} e^{-i\xi x} f(x) dx.$$

$\mathcal{K}_t$ on $\mathbb{R}^d$ is the heat operator, thus $\mathcal{K}_t^{1/2} = \mathcal{K}_{\frac{t}{2}}$. And

$$\mathcal{K}_t f(x) = \int_{\mathbb{R}^d} k_t(x,y) f(y) dy = (k_t * f)(x),$$

So, $F(\mathcal{K}_t f) = \hat{k}_t \hat{f}$. Note that $F$ is an isometry. Thus it is the same to transform the (35) using Fourier transform. Then we have

$$\min_{\hat{h} \in L_2} \left\| \hat{f} - \hat{k}_{\frac{t}{2}} \hat{h} \right\|_2^2 + \frac{\lambda'}{\Gamma^2} \|\hat{h}\|_2^2$$

where $\hat{k}_t(\xi) = e^{\frac{-\|\xi\|^2 t}{2}}$. And let

$$\tilde{f}(\xi) = \begin{cases} \hat{f}(\xi) & \text{if } \|\xi\|^2 < \frac{4}{t} \\ 0 & \text{Otherwise.} \end{cases}$$

and $\tilde{h} = \left( \hat{k}_{\frac{t}{2}} \right)^{-1} \tilde{f}$. It is obvious that $\|\tilde{h}\|_2^2 \leq e\|\hat{f}\|_2^2 = e\|f\|_2^2$. And

$$\left\| \hat{f} - \hat{k}_{\frac{t}{2}} \tilde{h} \right\|_2^2 = \left\| \hat{f} - \tilde{f} \right\|_2^2$$

Now we recall the definition of Sobolev space using Fourier transform, which states that $\hat{f}(\xi) = \frac{1}{(1+\|\xi\|^2)^{s/2}} \hat{u}(\xi)$ and $\|\hat{u}\|_2 \leq \|f\|_{W_s^2}$. Thus,

$$\left\| \hat{f} - \tilde{f} \right\|_2^2 = \left( \int_{\|\xi\|^2 \geq \frac{4}{t}} \frac{1}{(1+\|\xi\|^2)^{s/2}} \hat{u}(\xi) d\xi \right)^2 \leq \int_{\|\xi\|^2 \geq \frac{4}{t}} \left( \frac{1}{(1+\|\xi\|^2)^{s/2}} \hat{u}(\xi) \right)^2 d\xi \leq \frac{t^s}{4^s} \|f\|_{W_s^2}^2$$

Thus, we have

$$\min_{\hat{h} \in L_2} \left\| \hat{f} - \hat{k}_{\frac{t}{2}} \hat{h} \right\|_2^2 + \frac{\lambda'}{\Gamma^2} \|\hat{h}\|_2^2 \leq \left\| \hat{f} - \hat{k}_{\frac{t}{2}} \tilde{h} \right\|_2^2 + \frac{\lambda'}{\Gamma^2} \|\tilde{h}\|_2^2 \leq \frac{t^s}{4^s} \|f\|_{W_s^2}^2 + \frac{e\lambda'}{\Gamma^2} \|f\|_2^2$$

Let $D_1 = \frac{\Gamma^2}{4^s}$ and $D_2 = e\|f\|_2^2$, we have the lemma. $\qquad \square$

# B    Proof for Lemma 6

For the compact manifold case, we also need to have similar lemma as the above one. However, the definition of Fourier transform is obscure, thus we need to consider alternative way to get the same bound. We can use the Laplace-Beltrami operator on the compact manifold. It has discrete spectrum and satisfies the Weyl's Law, see Chapter 8 in [32], about the spectrum of the Laplace-Beltrami operator $\Delta$, which is discrete if the manifold is compact. It states the following: the number of eigenvalues of Laplacian operator over a bounded domain with Neumann Bounday condition that are less or equal than $x$, denoted by $N(x)$, satisfies

$$\lim_{x \to \infty} \frac{N(x)}{x^{d/2}} = C$$

for a constant $C$ depending on the dimensionality and volume of the domain. This implies there exists $M$ such that for any $i > M$,

$$c_1 i^{2/d} \leq \eta_i \leq c_2 i^{2/d}.$$

Also, we can redefine the Sobolev space on a compact manifold using Laplace-Beltrami operator.

$$W_s^2 = \{f \in L_2 : \|f\|_{W_s^2} = \sup_{v \leq s} \left\| \Delta^{v/2} f \right\|_2 < \infty\}.$$

And this definition of $W_2^s$ is equivalent to common definition of Sobolev space using differentiation, see [29] for the details for this equivalence.

First we need the following lemma.

**Lemma 10.** *Suppose $f \in W_2^s$ and $N_t = \left( \frac{1}{t} \right)^{\alpha}$, then we have*

$$\sum_{i > N_t} \langle f, v_i \rangle_2^2 \leq C \|f\|_{W_s^2}^2 t^{2\alpha s/d}$$

*where $v_i$ is the eigenfunctions of Laplacian operator $\Delta$ and $C$ is a constant independent of $t$.*

*Proof.* First let proof that

$$\sum_{i=0}^{\infty} \langle f, v_i \rangle_2^2 i^{2s/d} < \infty.$$

Using the implication of Weyl's Law, we have for $i > M$, $i^{2s/d} \leq \frac{\eta_i^s}{c_1}$. Thus,

$$
\begin{aligned}
\sum_{i=0}^{\infty} \langle f, v_i \rangle_2^2 i^{2s/d} &= \sum_{i \leq M}^{\infty} \langle f, v_i \rangle_2^2 i^{2s/d} + \sum_{i > M}^{\infty} \langle f, v_i \rangle_2^2 i^{2s/d} \\
&\leq \sum_{i \leq M}^{\infty} \langle f, v_i \rangle_2^2 i^{2s/d} + \sum_{i > M}^{\infty} \langle f, v_i \rangle_2^2 \frac{\eta_i^s}{c_1} \\
&\leq \sum_{i \leq M}^{\infty} \langle f, v_i \rangle_2^2 i^{2s/d} + \frac{1}{c_1} \left\| \Delta^{s/2} f \right\|_2^2 < C \|f\|_{W_s^2}^2
\end{aligned}
$$

For $N_t = \left( \frac{1}{t} \right)^{\alpha}$. We have

$$N_t^{2s/d} \sum_{i > N_t} \langle f, v_i \rangle_2^2 < \sum_{i > N_t} \langle f, v_i \rangle_2^2 i^{2s/d} \leq C \|f\|_{W_s^2}^2.$$

Thus,

$$\sum_{i > N_t} \langle f, v_i \rangle_2^2 < \frac{C \|f\|_{W_s^2}^2}{N_t^{2s/d}} = C \|f\|_{W_s^2}^2 t^{2\alpha s/d}.$$

$\square$

Now we can give the proof for Lemma 6.

*Proof.* RKHS is unique for a given domain and kernel, so is independent of the measure used to define the $L_{2,\rho}$. Thus for any $g \in L_{2,p}$, there should be $h \in L_2$ such that $\mathcal{L}_{t,p}^{1/2} g = \mathcal{L}_t^{1/2} h$ and

$$\|g\|_{2,p} = \|\mathcal{L}_{t,p}^{1/2} g\|_{H_t} = \|\mathcal{L}_t^{1/2} h\|_{H_t} = \|h\|_2.$$

Since this is true for arbitrary $g \in L_{2,p}$, we have

$$\min_{g \in L_{2,p}} \left\| f - \mathcal{L}_{t,p}^{1/2} g \right\|_{2,p}^2 + \lambda' \|g\|_{2,p}^2 = \min_{h \in L_2} \left\| f - \mathcal{L}_t^{1/2} h \right\|_{2,p}^2 + \lambda' \|h\|_2^2$$

Because $\| \cdot \|_{2,p} \le \Gamma \| \cdot \|_2$,

$$\min_{h \in L_2} \left\| f - \mathcal{L}_t^{1/2} h \right\|_{2,p}^2 + \lambda' \|h\|_2^2 \le \Gamma^2 \left( \min_{h \in L_2} \left\| f - \mathcal{L}_t^{1/2} h \right\|_2^2 + \frac{\lambda'}{\Gamma^2} \|h\|_2^2 \right)$$

Now, let

$$h_{\lambda'}^* = \arg \min_{h \in L_2} \left\| f - \mathcal{L}_t^{1/2} h \right\|_2^2 + \lambda' \|h\|_2^2$$

Expend $f$ using the eigenfunctions $v_0, v_1, \dots$ of $\Delta$, we have

$$f = \sum_{i=0}^{\infty} \langle f, v_i \rangle_2 v_i$$

Denote the eigenvalues of $\Delta$ as $\eta_0, \eta_1, \dots$, the heat operator defined as $H_t = e^{-\Delta t}$ having eigenvalues as $e^{-\eta_0 t}, e^{-\eta_1 t}, \dots$. Recall the Weyl's law, we have there exists $M$ such that for any $i > M$, $c_1 i^{2/d} \le \eta_i \le c_2 i^{2/d}$. When $t$ is small enough, we will have $N_t = \frac{1}{t^{d/2}} > M$. Since we order $\eta_i$ in non-decreasing order, for any $i < N_t$, we have $\eta_i \le \eta_{N_t} \le c_2 N_t^{2/d} = c_2/t$, also $e^{-\eta_i t} > e^{-c_2}$. Now denote $P_N$ be the operator that projects function $f \in L_2$ to the subspace spanned by first $N$ eigenfunctions of $\Delta$. Thus

$$P_{N_t} f = \sum_{i \le N_t} \langle f, v_i \rangle_2 v_i$$

where $v_i$ is the eigenfunction of $\Delta$. And let

$$\hat{h} = H_t^{-1/2} P_{N_t} f = \sum_{i=0}^{N_t} e^{\frac{\eta_i t}{2}} \langle f, v_i \rangle_2 v_i$$

Thus, we have

$$\arg \min_{h \in L_2} \left\| f - \mathcal{L}_t^{1/2} h \right\|_2^2 + \lambda' \|h\|_2^2 \le \left\| f - \mathcal{L}_t^{1/2} \hat{h} \right\|_2^2 + \lambda' \|\hat{h}\|_2^2$$

$$= \left\| \sum_{i > N_t} \langle f, v_i \rangle_2 v_i + \sum_{i \le N_t} \langle f, v_i \rangle_2 v_i - \mathcal{L}_t^{1/2} \hat{h} \right\|_2^2 + \lambda' \|\hat{h}\|_2^2$$

$$\le \left( \left\| \sum_{i > N_t} \langle f, v_i \rangle_2 v_i \right\|_2 + \left\| H^{1/2} H^{-1/2} P_{N_t} f - \mathcal{L}_t^{1/2} H^{-1/2} P_{N_t} f \right\|_2 \right)^2 + \lambda' \sum_{i=1}^{N_t} e^{\eta_i t} \langle f, v_i^t \rangle_2^2$$

$$\tag{36}$$

Now let us proceed by bound the above formula. By Lemma 10 with $N_t = \frac{1}{t^{d/2}}$ and $s = 2$, we have

$$\sum_{i=N_t+1}^{\infty} \langle f, v_i \rangle_2^2 \le C \|f\|_{W_s^2}^2 t^2$$

Also, for $i < N_t$, $e^{\eta_i t} \leq e^{c_2}$, thus $\|H^{-1/2} P_{N_t} f\|_2 \leq e^{c_2/2} \|P_{N_t} f\|_2 \leq e^{c_2/2} \|f\|_2$. Recall we have $\|\mathcal{H}_t^{1/2} - \mathcal{L}_t^{1/2}\|_{L_2 \to L_2} \leq C't$ for a constant $C'$. Thus, we have

$$\left\| H^{1/2} H^{-1/2} P_{N_t} f - \mathcal{L}_t^{1/2} H^{-1/2} P_{N_t} f \right\|_2 \leq C' e^{c_2/2} \|f\|_2 \, t \leq C'' e^{c_2/2} \|f\|_{W_s^2} \, t,$$

since $\|f\|_2$ can always be bounded by $\|f\|_{W_s^2}$ up to a constant.

For the third term in (36), we have

$$\lambda' \sum_{i=1}^{N_t} e^{\eta_i t} \langle f, v_i \rangle_2^2 \leq \lambda' e^{c_2} \sum_{i=1}^{N_t} \langle f, v_i \rangle_2^2 \leq \lambda' e^{c_2} \|f\|_2^2$$

Hence,

$$\left\| f - \mathcal{L}_{t,p}^{1/2} g_{\lambda'}^* \right\|_{2,p}^2 + \lambda' \|g_{\lambda'}^*\|_{2,p}^2 \leq \Gamma^2 \left( \sqrt{C} \|f\|_{W_s^2} \, t + C' e^{c_2/2} \|f\|_{W_s^2} \, t \right)^2 + \lambda' e^{c_2} \|f\|_2^2$$

When $t$ is small enough, $t^2 \leq t$, letting $D_1 = 2\Gamma^2(C + C'^2 e^{c_2})$, $D_2 = e^{c_2}$, we prove the lemma. $\square$

## C    Proof for Lemma 7

*Proof.* By definition of $\mathcal{K}_t$, we have $(\mathcal{K}_t f - f)(x) = (\mathcal{K}_t - \mathcal{I})f$. By results in [2], we have $(\mathcal{K}_t - \mathcal{I})f = t\Delta f + o(t)$ when $f$ is twice differentiable. Due to $f \in W_2^2(\mathbb{R}^d)$, we have $\|\Delta f\|_2 < \infty$. Thus, we have

$$\|\mathcal{K}_t f - f\|_{2,p} \leq \Gamma \|\mathcal{K}_t f - f\|_2 = \Gamma \|t\Delta f + o(t)\|_2 \leq \Gamma t \|\Delta f\|_2 + o(t) = O(t).$$

For manifold case, we have $(\mathcal{K}_t - \mathcal{D})f = t\Delta f + o(t)$, where $\mathcal{D}f = \int_\mathcal{M} k_t(x,y) dy f(x)$. Thus,

$$(\mathcal{K}_t - \mathcal{I})f = (\mathcal{K}_t - \mathcal{D})f + (\mathcal{D} - \mathcal{I})f.$$

For the first term, we have the same rate with $\mathbb{R}^d$. Now we proceed by bounding the second term.

$$\|(\mathcal{D} - \mathcal{I})f\|_2 = \left\| \left( \int_\mathcal{M} k_t(\cdot, y) dy - 1 \right) f(\cdot) \right\|_2 \leq \left\| \int_\mathcal{M} k_t(\cdot, y) dy - 1 \right\|_2 \|f\|_2$$

We know that $\|f\|_2 < \infty$.

Let $B_t(x) = \{ y \in \mathcal{M} : \|x - y\|_2 < t^{\frac{1}{2} - \varepsilon} \}$ and $R_t(x)$ is the projection of $B_t(x)$ on the $T_x \mathcal{M}$. In the following proof, we need to use change of variables to converting integral over a manifold to the integral over the tangent space at a specific point. For two points $x, y \in \mathcal{M}$, let $y' = \pi_x(y)$ be the projection of $y$ in the tangent space $T_x$ of $\mathcal{M}$ at $x$. Let $J_{\pi_x}|_y$ denote the Jacobian of the map $\pi_x$ at point $y \in \mathcal{M}$ and $J_{\pi_x^{-1}}\big|_{y'}$ is the inverse. For $y$ sufficiently close to $x$, we have

$$\|x - y\| = \|x - y'\| + O(\|x - y'\|^3)$$
$$\left| J_{\pi_x}|_y - 1 \right| = O(\|x - y\|^2)$$
$$\left| J_{\pi_x^{-1}}\big|_{y'} - 1 \right| = O(\|x - y'\|^2).$$

Thus, it is true that the points in $R_t(x)$ are still no further than $2t^{\frac{1}{2} - \varepsilon}$, when $t$ is small enough. Since $k_t$ has exponential decay, the integral $\int_{\overline{B_t(x)}} k_t(y, \cdot) dy$ is of order $O(e^{-t^{-\varepsilon}})$, and so is

$\int_{\overline{R_t(x)}} k_t(y', \cdot)dy'$. Thus, for any point $x \in \mathcal{M}$,

$$\left| \int_{\mathcal{M}} k_t(x, \cdot)dx - 1 \right|$$

$$= \left| \int_{B_t(x)} k_t(y, \cdot)dy - 1 + O(e^{-t^{-\varepsilon}}) \right|$$

$$= \left| \int_{B_t(x)} k_t(y, \cdot)dy - \int_{T_x\mathcal{M}} k_t(y', \cdot)dy' + O(e^{-t^{-\varepsilon}}) \right|$$

$$= \left| \int_{R(x)} k_t(y', \cdot)J_{\pi^{-1}}|_{y'}dy' - \int_{R(x)} k_t(x, \cdot)dx + O(e^{-t^{-\varepsilon}}) \right|$$

$$= \left| \int_{R(x)} k_t(x, \cdot)(J_{\pi^{-1}}|_x - 1)dx + O(e^{-t^{-\varepsilon}}) \right|$$

$$= O(t^{1-2\varepsilon}) \int_{R(x)} k_t(x, \cdot)dx + O(e^{-t^{-\varepsilon}})$$

$$= O(t^{1-2\varepsilon}) \left( \int_{T_x\mathcal{M}} k_t(x, \cdot)dx + O(e^{-t^{-\varepsilon}}) \right) + O(e^{-t^{-\varepsilon}})$$

$$= O(t^{1-2\varepsilon}) \left( 1 + O(e^{-t^{-\varepsilon}}) \right) + O(e^{-t^{-\varepsilon}})$$

$$= O(t^{1-2\varepsilon})$$

Abusing the notation of $\varepsilon$, we have $\| \int_{\mathcal{M}} k_t(\cdot, y)dy - 1 \|_2 \leq O(t^{1-\varepsilon})$ where $0 < \varepsilon < 1$. $\qquad\square$

## D  Proof for Theorems in 4.2.2

In the second case, since we do not have samples from $q$, we replace $\mathcal{K}_{t,q,\boldsymbol{z}_q}$ by $q$. Consider corresponding $f_\lambda^{\mathrm{II}}$,

$$f_\lambda^{\mathrm{II}} = (\mathcal{K}_{t,p}^3 + \lambda\mathcal{I})^{-1}\mathcal{K}_{t,p}^2 q = (\mathcal{K}_{t,p}^3 + \lambda\mathcal{I})^{-1}\mathcal{K}_{t,p}^2 \left( q - \mathcal{K}_{t,p}\frac{q}{p} + \mathcal{K}_{t,p}\frac{q}{p} \right)$$

$$= (\mathcal{K}_{t,p}^3 + \lambda\mathcal{I})^{-1}\mathcal{K}_{t,p}^2 \left( q - \mathcal{K}_{t,p}\frac{q}{p} \right) + (\mathcal{K}_{t,p}^3 + \lambda\mathcal{I})^{-1}\mathcal{K}_{t,p}^3 \frac{q}{p}$$

Thus, we need to bound the extra term $(\mathcal{K}_{t,p}^3 + \lambda\mathcal{I})^{-1}\mathcal{K}_{t,p}^2 \left( q - \mathcal{K}_{t,p}\frac{q}{p} \right)$. Let $d = q - \mathcal{K}_{t,p}\frac{q}{p}$ and $\|d\|_{2,p} = \delta_t$, we have

$$\left\| (\mathcal{L}_{t,p}^3 + \lambda\mathcal{I})^{-1}\mathcal{L}_{t,p}^2 d \right\|_{2,p} = \left( \sum_{i=1}^\infty \left( \frac{\sigma_i^2 \langle u_i, d\rangle_{2,p}}{\sigma_i^3 + \lambda} \right)^2 \right)^{\frac{1}{2}} \leq \max_{\sigma>0} \left( \frac{1}{\sigma + \frac{\lambda}{\sigma^2}} \right) \delta_t \leq \frac{\delta_t}{\left( 2^{\frac{1}{3}} + 2^{-\frac{2}{3}} \right) \lambda^{\frac{1}{3}}} \leq \lambda^{-\frac{1}{3}}\delta_t$$

The bound for $\delta_t$ is given in the following lemma.

For the concentration of $\|f_\lambda^{\mathrm{II}} - f_{\lambda,\boldsymbol{z}}^{\mathrm{II}}\|_{2,p}$, we will consider their close formulas

$$f_\lambda^{\mathrm{II}} = \left( \mathcal{K}_p^3 + \lambda\mathcal{I} \right)^{-1} \mathcal{K}_p^2 q$$

$$f_{\lambda,\boldsymbol{z}}^{\mathrm{II}} = \left( \mathcal{K}_{z_p}^3 + \lambda\mathcal{I} \right)^{-1} \mathcal{K}_{z_p}^2 q \tag{37}$$

By the similar argument to that in Lemma 9, we will have the following lemma gives the concentration bound.

**Lemma 11.** *Let $p$ be a density of a probability measure over a domain $X$ and $q$ another density function. They satisfy the assumptions in 4.1. Consider $f_\lambda^{II}$ and $f_{\lambda,\boldsymbol{z}}^{II}$ defined in Eq. 37, with confidence*

*at least* $1 - 2e^{-\tau}$, *we have*

$$\left\| f_\lambda^{II} - f_{\lambda,z}^{II} \right\|_{2,p} \le C_4 \left( \frac{\kappa_t \sqrt{\tau}}{\lambda^{3/2}\sqrt{n}} + \frac{\kappa_t \sqrt{\tau}}{\lambda\sqrt{n}} \right)$$

*where* $\kappa_t = \sup_{x \in \Omega} k_t(x,x) = \frac{1}{(2\pi t)^{d/2}}$

*Proof.* Let $\tilde{f} = \left( \mathcal{K}_{\mathbf{z}_p}^3 + \lambda \mathcal{I} \right)^{-1} \mathcal{K}_p^2 q$. We have $f_\lambda^{II} - f_{\lambda,z}^{II} = f_\lambda^{II} - \tilde{f} + \tilde{f} - f_{\lambda,z}^{II}$. For $f_\lambda^{II} - \tilde{f}$, using the fact that $\left( \mathcal{K}_p^3 + \lambda \mathcal{I} \right) f_\lambda^{II} = \mathcal{K}_p^2 q$, we have

$$
\begin{aligned}
& f_\lambda^{II} - \tilde{f} \\
={} & f_\lambda^{II} - \left( \mathcal{K}_{\mathbf{z}_p}^3 + \lambda \mathcal{I} \right)^{-1} \left( \mathcal{K}_p^3 + \lambda \mathcal{I} \right) f_\lambda^{II} \\
={} & \left( \mathcal{K}_{\mathbf{z}_p}^3 + \lambda \mathcal{I} \right)^{-1} \left( \mathcal{K}_{\mathbf{z}_p}^3 - \mathcal{K}_p^3 \right) f_\lambda^{II}
\end{aligned}
$$

And

$$
\begin{aligned}
& \tilde{f} - f_{\lambda,z}^{II} \\
={} & \left( \mathcal{K}_{\mathbf{z}_p}^3 + \lambda \mathcal{I} \right)^{-1} \mathcal{K}_p^2 q - \left( \mathcal{K}_{\mathbf{z}_p}^3 + \lambda \mathcal{I} \right)^{-1} \mathcal{K}_{\mathbf{z}_p}^2 q \\
={} & \left( \mathcal{K}_{\mathbf{z}_p}^3 + \lambda \mathcal{I} \right)^{-1} \left( \mathcal{K}_p^2 - \mathcal{K}_{\mathbf{z}_p}^2 \right) q
\end{aligned}
$$

Notice that we have $\mathcal{K}_{\mathbf{z}_p}^3 - \mathcal{K}_p^3$ and $\mathcal{K}_{\mathbf{z}_p}^2 - \mathcal{K}_p^2$ in the identity we get. For these two objects, it is not hard to verify the following identities,

$$
\begin{aligned}
& \mathcal{K}_{\mathbf{z}_p}^3 - \mathcal{K}_p^3 \\
={} & \left( \mathcal{K}_{\mathbf{z}_p} - \mathcal{K}_p \right)^3 + \mathcal{K}_p \left( \mathcal{K}_{\mathbf{z}_p} - \mathcal{K}_p \right)^2 + \left( \mathcal{K}_{\mathbf{z}_p} - \mathcal{K}_p \right) \mathcal{K}_p \left( \mathcal{K}_{\mathbf{z}_p} - \mathcal{K}_p \right) + \left( \mathcal{K}_{\mathbf{z}_p} - \mathcal{K}_p \right)^2 \mathcal{K}_p \\
& + \mathcal{K}_p^2 \left( \mathcal{K}_{\mathbf{z}_p} - \mathcal{K}_p \right) + \mathcal{K}_p \left( \mathcal{K}_{\mathbf{z}_p} - \mathcal{K}_p \right) \mathcal{K}_p + \left( \mathcal{K}_{\mathbf{z}_p} - \mathcal{K}_p \right) \mathcal{K}_p^2.
\end{aligned}
$$

And

$$\mathcal{K}_{\mathbf{z}_p}^2 - \mathcal{K}_p^2 = \left( \mathcal{K}_{\mathbf{z}_p} - \mathcal{K}_p \right)^2 + \mathcal{K}_p \left( \mathcal{K}_{\mathbf{z}_p} - \mathcal{K}_p \right) + \left( \mathcal{K}_{\mathbf{z}_p} - \mathcal{K}_p \right) \mathcal{K}_p$$

Thus, in these two identities, the only two random variables are $\mathcal{K}_{\mathbf{z}_p} - \mathcal{K}_p$. By results about concentration of $\mathcal{K}_{\mathbf{z}_p}$ and $\mathcal{K}_{\mathbf{z}_q}$, we have with probability $1 - 2e^{-\tau}$,

$$
\begin{aligned}
\left\| \mathcal{K}_{\mathbf{z}_p} - \mathcal{K}_p \right\|_{\mathcal{H} \to \mathcal{H}} &\le \frac{\kappa_t \sqrt{\tau}}{\sqrt{n}}, \\
\left\| \mathcal{K}_{\mathbf{z}_p} q - \mathcal{K}_p q \right\|_{\mathcal{H}} &\le \frac{\kappa_t \|q\|_\infty \sqrt{2\tau}}{\sqrt{n}}
\end{aligned}
\tag{38}
$$

And we know that for a large enough constant $c$ which is independent of $t$ and $\lambda$,

$$\left\| \mathcal{K}_p \right\|_{\mathcal{H} \to \mathcal{H}} < c, \left\| \left( \mathcal{K}_{\mathbf{z}_p}^3 + \lambda \mathcal{I} \right)^{-1} \right\|_{\mathcal{H} \to \mathcal{H}} \le \frac{1}{\lambda}, \left\| \mathcal{K}_p q \right\|_{\mathcal{H}} < c\|q\|_{2,p}$$

and

$$\left\| f_\lambda^{II} \right\|_{\mathcal{H}}^2 = \sum_i \frac{\sigma_i^3}{(\sigma_i^3 + \lambda)^2} \langle q, u_i \rangle^2 \le \left( \sup_{\sigma > 0} \frac{\sigma^3}{(\sigma^3 + \lambda)^2} \right) \sum_i \langle q, u_i \rangle^2 \le \frac{c^2}{\lambda} \|q\|_{2,p}^2$$

thus, $\left\| f_\lambda^{II} \right\|_{\mathcal{H}} \le \frac{c}{\lambda^{1/2}} \|q\|_{2,p}$.

Notice that $\left\| \left( \mathcal{K}_{\mathbf{z}_p} - \mathcal{K}_p \right)^2 \right\|_{\mathcal{H}} \le \left\| \mathcal{K}_{\mathbf{z}_p} - \mathcal{K}_p \right\|_{\mathcal{H}}^2$ and $\left\| \left( \mathcal{K}_{\mathbf{z}_p} - \mathcal{K}_p \right)^3 \right\|_{\mathcal{H}} \le \left\| \mathcal{K}_{\mathbf{z}_p} - \mathcal{K}_p \right\|_{\mathcal{H}}^3$, both of this could be of smaller order compared with $\left\| \mathcal{K}_{\mathbf{z}_p} - \mathcal{K}_p \right\|_{\mathcal{H}}$. For simplicity we hide the term including them in the final bound without changing the dominant order. We could also hide the

terms with the product of any two the random variables in Eq. 38, which is of prior order compared to the term with only one random variable. Now let us put everything together,

$$\|f_\lambda^{\mathrm{II}} - f_{\lambda,\boldsymbol{z}}^{\mathrm{II}}\|_{2,p} \leq c^{1/2}\|f_\lambda^{\mathrm{II}} - f_{\lambda,\boldsymbol{z}}^{\mathrm{II}}\|_{H_t}$$

$$\leq c^{1/2}\left(\frac{c^3 \kappa_t \sqrt{\tau}}{\lambda^{3/2}\sqrt{n}}\|q\|_{2,p} + \frac{c^2 \kappa_t \sqrt{\tau}}{\lambda\sqrt{n}}\|q\|_\infty\right)$$

$$\leq C_4 \left(\frac{\kappa_t \sqrt{\tau}}{\lambda^{3/2}\sqrt{n}} + \frac{\kappa_t \sqrt{\tau}}{\lambda\sqrt{n}}\right)$$

where $C_4 = c^{5/2}\max\left(c\,\|q\|_{2,p}, \|q\|_\infty\right)$. $\qquad\square$

Given the above lemmas, the main theorem for the second case follows.