[Reviews · NeurIPS 2013]

Submitted by Assigned_Reviewer_1

The paper proposes a new method to estimate the density ratio p/q. There are a couple of other methods already proposed to this problem. The novelty of this paper is that the authors reformulate the density ratio estimation as a Fredholm integral equation.

Strong points of the paper:

The paper is well written; I enjoyed reading it. The density ratio estimation problem is important and interesting.

Weak points of the paper, comments, and questions:

(i) Some details are missing from the paper. For example, why
\int k_t(x,y)f(x)dx=f(y)+ O(t)? Either a proof (in the paper or in the supplementary material) or a reference needed for all these kind of statements, even if they are easy to prove.

(ii) As a special case (q=1), this method can also be used to density estimation. It is interesting that this approach doesn’t require the bandwidth to converge to zero. The standard kde density estimation is not consistent in that case. It would be nice to see more discussion about this, so the readers could clearly see that the bandwidth parameters in the paper have different roles than that of in kde.

(iii) I recommend adding a few sentences about Tikhonov regularization. The majority of the NIPS community might be not familiar with it.

(iv) All the proofs are in the supplementary material. I haven’t checked if they are correct.

(v) It would be nice to read a few sentences about why a naive plug-in approach, that is separately estimate densities p and q, is worse than the method proposed in this paper.

(vi) It’s also not clear how this method performs compared to other methods. For example, if I remember correctly [12] derives minimax bounds too. Corollary 2 and 4 provide upper bounds on the convergence rate, but it is not clear how far they are from the optimal minimax rate.

(vii) Please provide references for the datasets used in the paper (CPUsmall, Kin8nm, USPS).

(viii) I missed the demonstration on how the proposed method would work on toy problems for some fixed and known p and q. For example, for 1 dimensional p and q, a plot of the true p/q and the estimated p/q could help the readers assess the quality of the estimation.

Summary: It’s a nice paper about density ratio estimation. It is not clear to me how this method would perform compared to other algorithms (e,g. [12]), and I missed the demonstration of how this approach would work on toy problems, e.g. using simple 1-dimensional p and q densities.

Submitted by Assigned_Reviewer_7

This paper discusses two new methods for estimating the density ratio of two probabilities. The first method uses the equation of importance sampling, and formulates it in the form of a regularization problem where L^2 norm of the deviation from the equality and RKHS regularization term are used (Type I Fredholm equation). The second method considers a kernel where the locality parameter should go to zero for consistency. The latter is also expressed in the form of regularization (Type II Fredholm equation). The paper shows some theoretical results on convergence, and demonstrates the practical performance experimentally.

While the method has similarity to existing methods such as KMM, it is a strong advantage of the proposed one that the L^2 formulation enables us to apply cross-validation (CV) approach to choose the parameters in the algorithm. Unlike supervised problems, where CV is dominant as a method for choosing parameters in practice, there are no general methods for unsupervised learning. The paper solves this issue for density ratio estimation with a clever idea of using L^2 norm.
It is nicer to make a remark that the objective function given by RKHS, such as in KMM, is not appropriate for choosing a kernel with CV, since the values of the objective function depend on the kernel, and thus not comparable.

In Theorem 1, by the (-t/log \lambda) factor, the derived convergence rate is slow with fixed t. In many literatures of kernel methods, with fixed t, one can derive a polynomial rate with respect to the sample size after fixing the rate of \lambda. Are there any reasonable additional assumptions that give a polynomial rate for the proposed estimator? Caponnetto and De Vito (2007), for example, make assumptions on the spectrum of the operator K_{p,t} to derive the optimal rate for kernel ridge regression. I guess that making a stronger assumption on q/p will also derive a polynomial rate. It will be more interesting if the authors discuss such polynomial rate under stronger assumptions with fixed t.

In estimating a density ratio, we usually need to make an assumption that q/p is in a nice function class such as Sobolev. It is important to note that this assumption is very strong, requiring some knowledge on the ratio or tails of p and q. Under such an assumption on q/p, the opposite ratio p/q is usually behaves badly. It is necessary to include some discussions somewhere on this limitation of density ratio estimation.

This is not a requirement, but the supplementary materials should be improved. There are many typos and minor mistakes, which make it hard to check the correctness of the theoretical results in the main paper. For instance, in Lemma 5, W_2^2 should be W_s^2, and the bound should be D_1 (t/-log \lambda)^2 + \lambda^{1-\alpha} D_2 \|f\|_2^2. The expressions at line 675 and 686 are incorrect, while Eq.(31) is correct. In the proof of Lemma 5, the domain of the integral at line 1147 must be \| \xi\|^2 \geq. Since the theoretical results of convergence consist of main contributions of this paper, I advise the authors to revise the supplementary material.


References:
Caponnetto A., De Vito E. Optimal Rates for Regularized Least-Squares Algorithm. Foundations of Computational Mathematics, 7 331-368 (2007).


Summary: The paper proposes a new approach to the density ratio estimation. The strong point is that CV can be effectively applied to choose parameters.

Submitted by Assigned_Reviewer_9

The authors propose a solution to the problem how to estimate the ratio of two probability density functions. The approach is based on using a kernel and a Fredholm integral equation. This yields
an inverse problem.
Summary: From the viewpoint of machine learning, the assumption of densities is a very hard one, because it is well-known that in any epsilon neighbourhood of a probability measure P which has a \nu-density there are probability measures which have no \nu-densities.

Therefore, the estimation of a ratio of two densities is problematic from the viewpoint of ML.
As far as I can see, the authors do not describe which dominating measure \nu is used (Lebesgue measure, counting measure, something else) and it is unclear to me, if their results are rigorous because of sets of \nu-measure zero. I also miss a clear statement which assumptions are made. E.g., does x belong to $R^d$ or to some general topological space? Is the density p positive for all x ?

Submitted by Assigned_Reviewer_10

The authors use two equalities to derive two estimators of the Radon-Nikodym derivative f= p/q:

(i) Ep ( k(x,.) f) = E_q k(x,.) and (ii) Ep (k(x,.) f ) ~ q(x)

The integral operators are replaced with empirical versions. {k(x,.)} can be seen as a set of test functions over which the difference between the right and left side is minimised. The choice of {k(x,.)} as the family of test functions has the advantage that standard kernel methods can be used for optimisation.

Convergence guarantees are given for both settings under smoothness & boundedness assumptions on the densities. The rate of convergence depends crucially on the dimension of the underlying space -- as one would expect for a density estimate.

I think it is nice and well executed work.

A couple of comments:

- For the convergence rates. Optimising over the set {k(x,.)} is sufficient to guarantee convergence to the ratio. For (ii) this is intuitive given the bandwidth dependence. Would be nice to have a short intuitive argument why this also works out in (i).

-You optimise on page 2 over L_2,p. Later in (9) you have a combination of L_2,p and L_2,q. Some motivation for the choice of cost function would be nice. I guess ultimately these need to stem from the tasks one is interested in. So would be nice to link the cost function choice to certain important applications.

- Speaking of applications, one example you got is importance sampling. Ie q/p is relevant for this to move from Eq to Ep. I'm wondering here: you are effectively estimating E_p and E_q to get your empirical cost function. So why not use directly the E_q you got there? Can E_p (q/p) be any better than directly E_q? I guess I'm asking here for a motivation of the setup since you need to produce estimates E_p and E_q to estimate q/p. You have a sentence on p 5 about it. That is sampling is difficult from E_q. Is this the kind of main application you have in mind?

- The regulariser: You enforce smoothness for the ratio q/p. Later in the theorem you got smoothness assumptions on q and p ( which I would say is the natural thing). So I'm wondering if there is any sort of relation between smoothness on q and p and smoothness of the ratio? I guess an answer is here already that you have many different q and p which produce the same ratio; ie just change on countable or on uncountable many points q and p to make it extremely irregular. So it might be more p ,q smooth (possibly bounded away from 0) => ratio smooth (?) or ratio smooth => there exist a smooth p and q which produces the ratio.

- Would be nice to discuss the d dependence of your rates -- for what dimensions would you expect your method to be useful?

- Is there any good motivation for your family of test functions U?

- A more principled question: For this q/p ; covariate shift etc setting one of the main difficulties seems to be to come up with a generic cost function. One approach is to go through expectation operators and throw different test functions at these to extract the underlying densities. Do you feel that you got here "the right" approach with the {k(x,.)} ? There is also some similarity to [1] in terms of playing around with transformations to get hold of the quantities of interest. Certainly, your approach for the Radon Nikodym derivative is nicer by working with test functions. But the overall approach has some similarity.


[1] Smooth Operators; Grunewalder, Gretton, Shawe-Taylor
Summary: I think it is a timely and strong paper addressing a relevant problem, providing a convergence study and experimental results.
Author Feedback

Author rebuttal: We thank the reviewers for many insightful comments, suggestions and corrections. They will be very useful for preparing the next revision of the paper.

Below we will discuss some of the questions/comments.

Reviewer_1:

Connection to density estimation:

This is a very good point. There are some interesting connections to both kernel density estimation and one-class SVM type methods.
In fact, we were initially planning to have a more in-depth discussion of these connections, but ultimately did not do that because of the NIPS space limits.

Comparison to [12] and other methods.

You are quite right that [12] establishes minimax rates for likelihood ratio estimation.
The direct comparison is somewhat difficult, since [12] uses the Hellinger metric and quite different methodology. There is also a recent theoretical analysis of KMM in [24]. However, the bounds there are not for the density ratio function as in our paper, but for the output of the corresponding integral operators. We also provide polynomial rates for Sobolev spaces, when the density ratio is not in the RKHS.

We will clarify this in the revision.


Reviewer_10.

Directly estimating $E_q$:
In the semi-supervised setting one may not have any (or enough) labeled data (from $q$) for estimating the loss of a classifier w.r.t. $q$. On the other hand, given sufficient amount of unlabeled data, the ratio of the marginals $q_X/p_X$ can be accurately estimated. In the MCMC setting the main issue is the computational difficulty of sampling from $q$, in some cases $p$ may be easier to sample from, while the ratio $q/p$ may be estimated analytically. We are still in the process of exploring that setting.


"-The regularizer".

Good point. Right now, we assume that q and p/q are smooth (in a certain function class). Perhaps, there is a way to put conditions on the ratio directly. That could also help generalize our result to the more general Radon-Nikodym derivative setting.

"-A more principled question" -- Thanks for the reference. Certainly, we would not go so far as to say that ours is "the right approach",
but we would argue that this is one natural way to address the problem. We feel that using our kernel framework with test functions
for parameter selection has some advantages over directly using a set of test functions as a basis (for example, as in LSIF, [9]).
It makes theoretical results easier to obtain and also allows us to use very restricted classes of functions (e.g., linear) for model selection,
which may not be very useful as a basis.

It would definitely be interesting and should be possible to prove some optimality results for our model selection procedure, but we do not have such results yet.


Reviewer_7.

Convergence rate:

For the Gaussian with fixed kernel width, the logarithmic rate of convergence seems unavoidable. That relates to the fact that the decay of the kernel spectrum is exponential, while the decay of coefficients for a function in a Sobolev space
is generally only polynomial. However, this can be overcome and a polynomial convergence rate can be obtained by choosing the width of the kernel adaptively (Corollary 2 and 4). Interestingly, as also noted by Reviewer 1, adaptive kernel parameter is necessary for the usual kernel density estimation procedures, but only desirable in our setting.

Alternatively, as done in Caponetto, De Vito, 07 (in a slightly different setting) one can control the spectral decay rate of the kernel by choosing, e.g.,
the Laplacian kernel whose spectral decay rate is polynomial. That type of analysis can definitely be done in our setting. The reason we concentrate on the Gaussian kernel is the connection to the practical implementations, where they are used more frequently than other types of kernels.

The Sobolev assumption:

Thanks for bringing up this point. This is quite a subtle issue. We actually don't view the Sobolev assumption as very strong. For example, $p$ and $q$ can be zero outside of a bounded domain (as long as the ratio is properly defined). In fact, we have some results for domains with boundary, which are briefly mentioned in the long version.
Still, a better way to weaken the conditions on the tail behavior would be to consider the Sobolev space with respect to the measure $p$, rather than the standard Sobolev space. It seems that our arguments should still go through, but we have not done a complete analysis yet.

Also thanks for the corrections.


Reviewer_9.

We are somewhat puzzled by the comments. It would certainly be impossible to provide theoretical results without some restrictions on the class of measures we are dealing with. The existence of density is a very common assumption in the extensive literature on density estimation and related areas.
We consider the two main cases, when the density is defined on $R^d$ or on a submanifold of $R^d$ (the dominating measure being the standard measure on $R^d$ and the uniform measure on the submanifold respectively). Both are stated in the paper.
To the best of our knowledge this is the first theoretical result for ratio estimation in the second setting.